# Fast weight programming and linear transformers: from machine learning to neurobiology

**Kazuki Irie**[*]  *kirie@fas.harvard.edu*
*Department of Psychology and Center for Brain Science*
*Harvard University, Cambridge, MA, USA*

**Samuel J. Gershman**[*]  *gershman@fas.harvard.edu*
*Department of Psychology and Center for Brain Science*
*Kempner Institute for the Study of Natural and Artificial Intelligence*
*Harvard University, Cambridge, MA, USA*

**Reviewed on OpenReview:** *https://openreview.net/forum?id=TDG8EkNmQR*

## Abstract

Recent advances in artificial neural networks for machine learning, and language modeling in particular, have established a family of recurrent neural network (RNN) architectures that, unlike conventional RNNs with vector-form hidden states, use two-dimensional (2D) matrix-form hidden states. Such 2D-state RNNs, known as Fast Weight Programmers (FWPs), can be interpreted as a neural network whose synaptic weights (called *fast weights*) dynamically change over time as a function of input observations, and serve as short-term memory storage; corresponding synaptic weight modifications are controlled or *programmed* by another network (the programmer) whose parameters are trained (e.g., by gradient descent). In this Primer, we review the technical foundations of FWPs, their computational characteristics, and their connections to transformers and state space models. We also discuss connections between FWPs and models of synaptic plasticity in the brain, suggesting a convergence of natural and artificial intelligence.

## 1 Introduction

While early development of artificial neural networks (ANNs) was loosely inspired by neuroscience (McCulloch and Pitts, 1943; Rosenblatt, 1958; Fukushima, 1980), ANNs have rapidly evolved into an independent subfield of machine learning (ML) and artificial intelligence (AI) on their own (Rumelhart et al., 1986b; McClelland et al., 1986; Ivakhnenko, 1971; LeCun et al., 2015; Schmidhuber, 2015). While some argue for the continued influence of neuroscience on the development of ANNs (Zador et al., 2023; Hassabis et al., 2017; Macpherson et al., 2021), in reality, progress in ANNs has been largely driven by the core pursuits of computer scientists to develop ever more powerful, general, and efficient ML models, rather than through their original motivation to model brain computation under neurobiological constraints (Gershman, 2024). In a twist of fate, such a detachment of the ANN research from the traditional goals of neuroscience has led to more open-ended and flourishing developments in ANN models, which, in turn, have attracted interest from cognitive neuroscientists, as they represent the best existing computational systems for processing vision, audio, and language (Achiam et al., 2023; Pratap et al., 2024)—modalities that are central to human cognition in real-life scenarios. Such successes of ANNs have stimulated many cognitive neuroscientists to seriously examine ML-driven models as hypotheses to explain neural computation in the brain (Kriegeskorte, 2015; Yamins and DiCarlo, 2016; Schrimpf et al., 2021; Gershman et al., 2025).

However, given the rapid progress in ML, there is still a significant gap between the computational modeling toolkits in the two fields. In particular, boosted by the recent successes of ChatGPT (Achiam et al., 2023)

---

[*]Kazuki Irie and Sam Gershman took the lead in writing the machine learning and neuroscience sections, respectively.

and other language models, a myriad of *sequence processing neural network* architectures have been proposed to improve upon the standard transformer architecture (reviewed later) (Vaswani et al., 2017). While keeping track of every such models has become particularly challenging today, as the lack of formal naming conventions makes their (often obvious) mathematical relations opaque (e.g., how does "Mamba2" (Dao and Gu, 2024) relate to "Gated Linear Attention" (Yang et al., 2024a)?; answered in Sec. 3.4), it seems useful to summarize key elements of such advances in sequence and memory modeling in ML that are arguably relevant to neuroscience.

In this Primer, we present a special family of recurrent neural networks (RNNs; Elman (1989); Jordan (1986)) called Fast Weight Programmers (FWPs; Schmidhuber (1992b); Schlag et al. (2021a); Irie et al. (2021)), that has been well established in machine learning, but has yet to see broad dissemination and application within the neuroscience community. Unlike the conventional RNN with one-dimensional vector-form hidden states, states in FWPs are two-dimensional (2D) matrices (see Figure 1 for illustrations). As we will argue, such 2D-state RNNs are particularly relevant for neuroscience, as the matrix-form states can be interpreted as time-varying synaptic weights that maintain short-term memory—unlike conventional RNNs, in which all the synaptic weights are fixed after training.

Additionally, FWPs naturally introduce a novel perspective on achieving biologically compatible local learning in ANNs, with an intuitive connection to the now-popular ML concept of *in-context learning*. Overall, FWPs may address certain longstanding limitations of ANNs as models of their biological counterpart, by providing a novel timescale for learning and memory.

The FWP concept is an entry point for understanding a multitude of sequence models recently proposed in machine learning. In fact, many such models can be directly expressed as an instantiation of FWPs, with a specific choice of the update rule used to modify the fast synaptic weights (see Table 1 for a preview); and we also review a formal connection between FWP and the transformer architecture (Vaswani et al., 2017).

We hope this Primer will stimulate neuroscientists to rethink computational modeling of certain neurobiological features in ANNs, through unique properties of FWPs; and help them familialize with the state-of-the-art sequence and memory models from machine learning.

> **Glossary (Machine Learning)**
>
> - **Efficient sequence model:** a parameterized sequence model whose training can be parallelized over the sequence length, and whose inference time-complexity is linear in sequence length.
> - **In-context learning:** an ability of a sequence model to learn a new task when a sequence of task demonstrations is fed to its input.
> - **Metalearning:** a process of leveraging learning experiences to acquire or improve the ability to learn.
> - **Model expressivity:** the range of computations the model can perform.
> - **Sequence processing neural networks:** a type of artificial neural network designed to handle a sequence of inputs.

## 2 Preliminaries

Before delving into fast weight programmers (FWPs) in the next section, we briefly review some of the conventional, general-purpose sequence processing neural networks: the conventional RNN (Elman, 1989; 1990) (Sec. 2.1) and related state space models (Sec. 2.2), and the transformer neural network (Vaswani et al., 2017) (Sec. 2.3). This will be useful later for contrasting, relating and characterizing properties of FWPs compared to these conventional sequence models.

Note that our main focus here is on sequence processing RNNs, rather than other "non-sequential RNNs" such as Amari-Little-Hopfield networks (Amari, 1972; Little, 1974; Hopfield, 1982). We also assume that readers are familiar with the general idea of machine learning that "trainable parameters" of a model can be trained by using some learning algorithm (e.g., backpropagation through time; BPTT (Rumelhart et al., 1986a; Werbos, 1990)) given some dataset or environment and an objective function to be optimized.

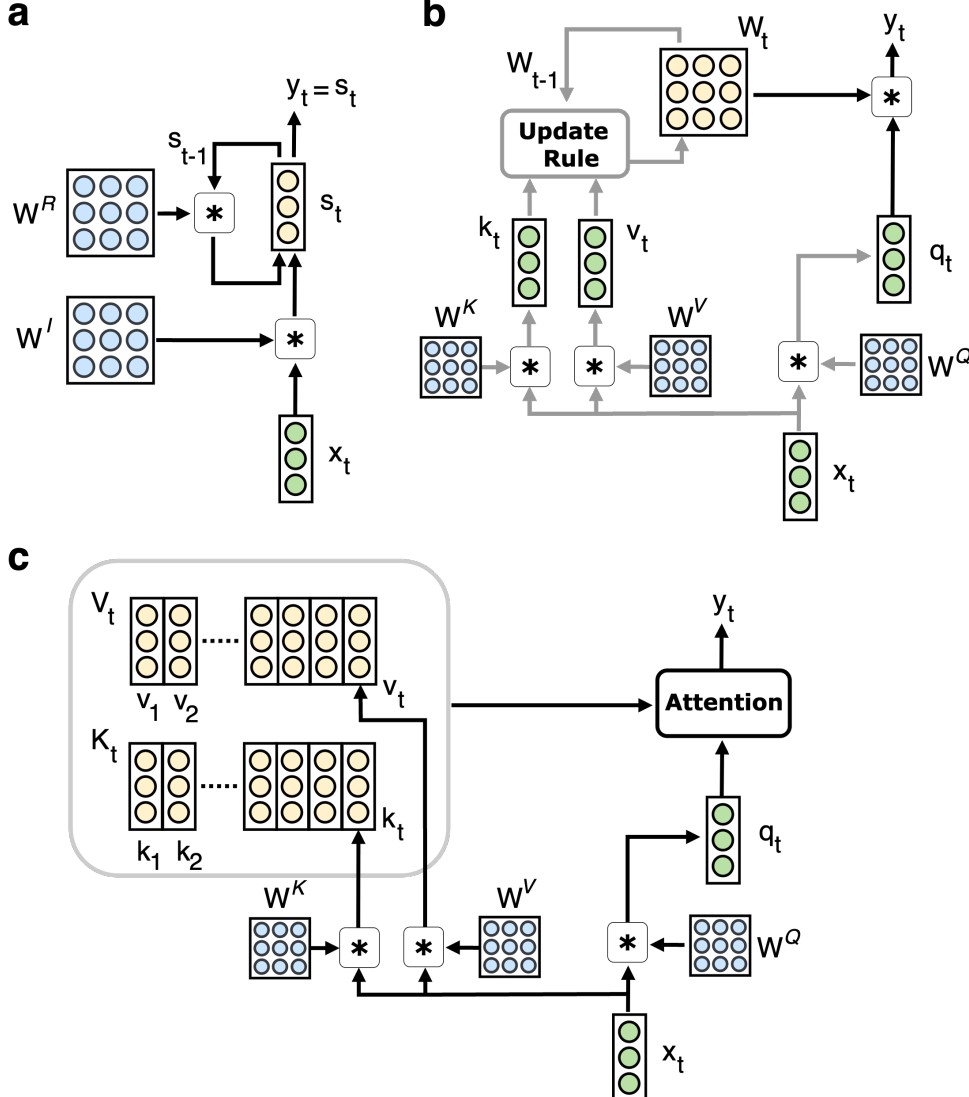

Figure 1: An Illustration of sequence processing in **a:** conventional recurrent neural networks (RNNs), **b:** fast weight programmers (FWPs), and **c:** transformers. One time step of recurrent computation is shown. In all figures, **colored circles** indicate time-step specific variables (**green**) that are not retained for the next time step, temporally changing model state/short-term memory (**yellow**), and model parameters that are fixed/frozen after training (**blue**). In particular, the hidden state $W_t$ of an FWP is a context-dependent time-varying matrix, whereas the hidden state of a conventional RNN $s_t$ is a vector, and that of a transformer is the key-value memory matrices $K_t$ and $V_t$ whose size grows linearly with the sequence length. In **b**, **black arrows** indicate computation in the fast/main net of the FWP, while the remaining **gray arrows** correspond to the computation performed by the slow/programmer net. Activation functions and variables that are specific to certain variants of FWPs (such as a dynamic learning rate or state decay factor) are omitted for clarify; see Table 1 for a specific choice of the update rule used in various models.

Throughout our Primer, $t$, $\tau$, $T$, $H$, $d$, $d_{\text{key}}$, $d_{\text{in}}$ and $d_{\text{out}}$ denote positive integers; $\odot$ and $\otimes$ denote element-wise multiplication and outer product, respectively. Our vectors are column vectors, which implies that outer product also writes as $\boldsymbol{a} \otimes \boldsymbol{b} = \boldsymbol{a}\boldsymbol{b}^{\top} \in \mathbb{R}^{n \times m}$ for arbitrary vectors $\boldsymbol{a} \in \mathbb{R}^n$ and $\boldsymbol{b} \in \mathbb{R}^m$ (this remark is useful to recognize outer products in certain equations).

## 2.1 Conventional recurrent neural networks

At every time step $t$, the conventional sequence processing RNN with a hidden state $\boldsymbol{s}_{t-1} \in \mathbb{R}^{d_{\text{out}}}$, receives an input $\boldsymbol{x}_t \in \mathbb{R}^{d_{\text{in}}}$ and produces an output $\boldsymbol{s}_t \in \mathbb{R}^{d_{\text{out}}}$ as follows:

$$\boldsymbol{s}_t = \sigma(\boldsymbol{W}^R \boldsymbol{s}_{t-1} + \boldsymbol{W}^I \boldsymbol{x}_t) \tag{1}$$

where $\sigma$ is an activation function (e.g., tanh), and $\boldsymbol{W}^R \in \mathbb{R}^{d_{\text{out}} \times d_{\text{out}}}$ and $\boldsymbol{W}^I \in \mathbb{R}^{d_{\text{out}} \times d_{\text{in}}}$ are recurrent and input weight matrices, respectively, which are the trainable parameters of the model. The initial state $\boldsymbol{s}_0$ is typically set to the "zero vector" whose entries are all zero. We omit the additive bias term inside $\sigma$ which is irrelevant for our discussion. See Figure 1a for an illustration.

Note that, in machine learning, this RNN architecture is never used as-is in practice today, as it is known to critically suffer from the classic "vanishing gradient problem" (Hochreiter, 1991; Bengio et al., 1994; Hochreiter et al., 2001a) when trained using a gradient-descent based learning algorithm, and to yield sub-optimal performance in practice (note that another well-known problem, the "exploding gradient problem" can be rather easily remediated by heuristically clipping/truncating large gradients to a certain value (Graves, 2013; Mikolov, 2012)). Instead, more sophisticated "gated architectures" such as long short-term memory (LSTM (Hochreiter and Schmidhuber, 1997; Gers et al., 2000; Greff et al., 2016)) are typically used. The core temporal dynamics of LSTMs maintains two recurrent states $\boldsymbol{c}_t \in \mathbb{R}^{d_{\text{out}}}$ and $\boldsymbol{s}_t \in \mathbb{R}^{d_{\text{out}}}$:

$$\boldsymbol{c}_t = \boldsymbol{r}_t \odot \boldsymbol{c}_{t-1} + \boldsymbol{i}_t \odot \boldsymbol{z}_t \tag{2}$$

$$\boldsymbol{s}_t = \boldsymbol{o}_t \odot \tanh(\boldsymbol{c}_t) \tag{3}$$

where the "gate functions" $\boldsymbol{r}_t, \boldsymbol{i}_t, \boldsymbol{o}_t \in \mathbb{R}^{d_{\text{out}}}$ and "cell input" $\boldsymbol{z}_t \in \mathbb{R}^{d_{\text{out}}}$ ($\boldsymbol{c}_t$ is often referred to as "cell state") are all parameterized functions of the recurrent state $\boldsymbol{s}_{t-1}$ and an input $\boldsymbol{x}_t$ as in Eq. 1.

Nevertheless, the vanilla RNN with the above Eq. 1 is sufficient as a contrastive example to later highlight the unique properties of FWPs. In this RNN model, during training, the "synaptic weights" $\boldsymbol{W}^R$ and $\boldsymbol{W}^I$ are modulated by the learning algorithm (typically by using gradients computed through backpropagation through time); however, once the training ends, these weights become frozen and immutable. At test time, the state vector $\boldsymbol{s}_t$ is the only time-varying variables which carry the model's short-term memory to process sequence elements over time. Later, we will show how FWPs critically differ from the conventional RNN on this aspect.

## 2.2 State Space Models

Given the current ML research trend (Tiezzi et al., 2025) of developing *efficient sequence models*—defined as models for which training can be parallelized over the time dimension, whereas inference time-complexity is linear in sequence length (Yau et al., 2025), the so-called "state space models (SSMs)" or "linear RNNs" have become popular (Gu et al., 2022; 2021). An SSM can be directly obtained from the conventional RNN above by simply removing the (non-linear) activation function $\sigma$ in Eq. 1:

$$\boldsymbol{s}_t = \boldsymbol{W}^R \boldsymbol{s}_{t-1} + \boldsymbol{W}^I \boldsymbol{x}_t \tag{4}$$

While deriving such a model from the original RNN is straightforward, this model is also known as a "linear time-invariant dynamical system" (which is originally defined through a continuous-time differential equation, whose discretization yields Eq. 4 exactly, which is typically followed by an extra equation $\boldsymbol{y}_t = \boldsymbol{W}^O \boldsymbol{s}_t$ to produce the output vector $\boldsymbol{y}_t \in \mathbb{R}^{d_{\text{out}}}$ using an extra weight matrix $\boldsymbol{W}^O \in \mathbb{R}^{d_{\text{out}} \times d_{\text{out}}}$). We do not further delve into such an alternative view here, as it provides no additional insight into the resulting computational model for our purpose.

The definition of an SSM can be generalized to include models with time-varying weight matrices $\boldsymbol{W}_t^R \in \mathbb{R}^{d_{\text{out}} \times d_{\text{out}}}$ and $\boldsymbol{W}_t^I \in \mathbb{R}^{d_{\text{out}} \times d_{\text{in}}}$ indexed by $t$:

$$\boldsymbol{s}_t = \boldsymbol{W}_t^R \boldsymbol{s}_{t-1} + \boldsymbol{W}_t^I \boldsymbol{x}_t \tag{5}$$

By recognizing that such a definition can also express models in which $\boldsymbol{s}_t$ is a matrix instead of a vector, we will see a connection to fast weight programmers (Sec. 3).

An interesting class of SSMs can be obtained by setting the weight matrices $\boldsymbol{W}_t^R$ and $\boldsymbol{W}_t^I$ in Eq. 5 to be diagonal matrices with diagonals $\boldsymbol{r}_t \in \mathbb{R}^{d_{\text{out}}}$ and $\boldsymbol{i}_t \in \mathbb{R}^{d_{\text{out}}}$ (by setting $d_{\text{out}} = d_{\text{in}}$), respectively; we obtain the following element-wise recurrence:

$$\boldsymbol{s}_t = \boldsymbol{r}_t \odot \boldsymbol{s}_{t-1} + \boldsymbol{i}_t \odot \boldsymbol{x}_t \tag{6}$$

It should be noted that this equation is identical to the cell update in an LSTM (Eq. 2), except that, in this SSM, $\boldsymbol{r}_t$ and $\boldsymbol{i}_t$ are functions of $\boldsymbol{x}_t$ or a few earlier observations (e.g., the last four; $\boldsymbol{x}_t$, $\boldsymbol{x}_{t-1}$, $\boldsymbol{x}_{t-2}$, $\boldsymbol{x}_{t-3}$) only, i.e., the "gate functions" are not recurrent (no dependency on $\boldsymbol{s}_{t-1}$), and the cell input $\boldsymbol{z}_t$ in Eq. 2 is reduced to $\boldsymbol{x}_t$.

An efficient SSM can be obtained by deliberately using this linear element-wise recurrence of Eq. 6 as the core temporal processing operation of the model (no other form of recurrence is used). This choice has two crucial consequences: first, it enables training parallelism (i.e., there exists an efficient algorithm to compute $\boldsymbol{s}_t$ for all $t$ in parallel (Martin and Cundy, 2018; Blelloch, 1990); which is not the case for the conventional, fully recurrent network); second, it sacrifices *model expressivity*, i.e., abilities to perform certain computations (Merrill et al., 2020; Grazzi et al., 2025; Merrill et al., 2024) (we discuss this further in Sec. 3.6)—as a side note, there is also a third consequence, which is that element-wise recurrence enables efficient online learning through a tractable real-time recurrent learning algorithm (Mozer, 1989; Gori et al., 1989; Zucchet et al., 2023; Irie et al., 2024), but further discussion is out of scope here.[1]

This "cell-only LSTM" is currently often called "Mamba", after the name of an efficient "hardware-aware implementation" (Gu and Dao, 2024), i.e., an implementation that takes into account the low-level memory hierarchy of modern graphic processing units (GPUs). However, we note that many have previously proposed similar models based on Eq. 6 under various names—including Quasi RNNs (Bradbury et al., 2017) and the Simple Recurrent Unit (Lei et al., 2018), among others (Balduzzi and Ghifary, 2016; Li et al., 2018; Gonnet and Deselaers, 2020)—to achieve improved efficiency over LSTM.

## 2.3 Transformer neural networks

Here we review the Transformer neural network (Vaswani et al., 2017) which is today's *de facto* standard sequence processing model architecture in ML. It also has a direct mathematical connection to FWPs (Schmidhuber, 1992b; Schlag et al., 2021a), as we will review in the next section.

Transformer models come in three distinct variations: encoder-decoder (Vaswani et al., 2017), encoder-only (Devlin et al., 2019), or decoder-only (Liu et al., 2018) architectures. We focus on the decoder-only (also known as the "causal model" architecture), which is a general-purpose sequence model used for example in language models (Al-Rfou et al., 2019; Dai et al., 2019; Baevski and Auli, 2019; Irie et al., 2019) including OpenAI's GPT series (Radford et al., 2019; Brown et al., 2020); for brevity, we refer to this as "the" transformer architecture here.

A typical transformer architecture consists of multiple layers, interleaving a "self-attention" layer (Parikh et al., 2016; Cheng et al., 2016; Lin et al., 2017; Bahdanau et al., 2015) and a two-layer feedforward block, each combined with a residual connection (He et al., 2016b;a; Srivastava et al., 2015) and a layer normalization operation (Ba et al., 2016b). Among these layers, the core temporal/memory processing operation in the transformer is carried out by the self-attention layer; therefore, we focus on describing its sequential dynamics here.

Like the conventional RNN in Sec. 2.1, at every time step $t$, a causal self-attention layer receives an input $\boldsymbol{x}_t \in \mathbb{R}^{d_{\text{in}}}$ and produces an output $\boldsymbol{y}_t \in \mathbb{R}^{d_{\text{out}}}$, while maintaining the so-called "key-value memory", represented

---

[1]As a further aside, in theory the expressivity of such element-wise RNNs could be enhanced by using complex-valued diagonals in the recurrent weight matrix (Orvieto et al., 2023; Ran-Milo et al., 2024) (which yields expressivity of a real-valued full matrix); however, stable implementation thereof is often reported to be challenging in practice (Elelimy et al., 2024).

by two matrices $\boldsymbol{K}_t \in \mathbb{R}^{d_{\text{key}} \times t}$ and $\boldsymbol{V}_t \in \mathbb{R}^{d_{\text{out}} \times t}$ as follows:

$$\boldsymbol{q}_t = \boldsymbol{W}^Q \boldsymbol{x}_t \ ; \ \boldsymbol{k}_t = \boldsymbol{W}^K \boldsymbol{x}_t \ ; \ \boldsymbol{v}_t = \boldsymbol{W}^V \boldsymbol{x}_t \tag{7}$$

$$\boldsymbol{K}_t = [\boldsymbol{K}_{t-1}, \boldsymbol{k}_t] \ ; \ \boldsymbol{V}_t = [\boldsymbol{V}_{t-1}, \boldsymbol{v}_t] \tag{8}$$

$$\boldsymbol{y}_t = \text{Attention}(\boldsymbol{K}_t, \boldsymbol{V}_t, \boldsymbol{q}_t) = \boldsymbol{V}_t \text{softmax}(\boldsymbol{K}_t^\top \boldsymbol{q}_t) \tag{9}$$

$$= \sum_{\tau=1}^{t} \boldsymbol{\alpha}_{t,\tau} \boldsymbol{v}_\tau \tag{10}$$

where $\boldsymbol{q}_t, \boldsymbol{k}_t \in \mathbb{R}^{d_{\text{key}}}$, $\boldsymbol{v}_t \in \mathbb{R}^{d_{\text{out}}}$ in Eq. 7 are different projections of the input—called query, key, and value, respectively— through matrices $\boldsymbol{W}^{\text{Q}} \in \mathbb{R}^{d_{\text{key}} \times d_{\text{in}}}$, $\boldsymbol{W}^{\text{K}} \in \mathbb{R}^{d_{\text{key}} \times d_{\text{in}}}$, and $\boldsymbol{W}^{\text{V}} \in \mathbb{R}^{d_{\text{out}} \times d_{\text{in}}}$ which are the trainable parameters of the model. The operation [] as in $[\boldsymbol{K}_{t-1}, \boldsymbol{k}_t]$ denotes concatenation of vector $\boldsymbol{k}_t \in \mathbb{R}^{d_{\text{key}}}$ to matrix $\boldsymbol{K}_{t-1} \in \mathbb{R}^{d_{\text{key}} \times (t-1)}$ along the time dimension, yielding $\boldsymbol{K}_t \in \mathbb{R}^{d_{\text{key}} \times t}$. $\boldsymbol{K}_0$ and $\boldsymbol{V}_0$ are initially empty. We omit the $1/\sqrt{d_{\text{key}}}$ scaling inside softmax, as well as the output projection, which are typically used but are irrelevant to our discussion here. In Eq. 10, $\boldsymbol{\alpha}_{t,\tau} \in \mathbb{R}$ denotes the $\tau$-th element of the vector $\text{softmax}(\boldsymbol{K}_t^\top \boldsymbol{q}_t) \in \mathbb{R}^t$. See Figure 1c for illustration.

The computation in Eq. 9 is called "attention", which is rather an intuitive name when we read Eq. 9 as (1) comparing the query vector from the current step $t$ to the keys from all the time steps through dot product to produce a score for each of $t$ keys ($\boldsymbol{K}_t^\top \boldsymbol{q}_t \in \mathbb{R}^t$), (2) sharpening and normalizing these similarity scores through the softmax function to obtain "attention scores" ($\boldsymbol{\alpha}_{t,\tau} \geq 0$ for all $\tau$ from 1 to $t$ with $\sum_{\tau=1}^{t} \boldsymbol{\alpha}_{t,\tau} = 1$), and (3) using the resulting scores as coefficients to compute the weighted average of all the value vectors (as is explicitly shown in Eq. 10) to produce the output; this effectively implements a form of attention as the model focuses on certain key-value memory elements at each step $t$.[2]

Additionally, practical transformers typically make use of "multi-head self-attention", where multiple independent heads within the same layer perform self-attention in parallel. By introducing an extra hyper-parameter $H$ as the number of heads, after projection of Eq. 7, each query/key/value vector is split into $H$ sub-vectors of the same size ($d_{\text{key}}$ and $d_{\text{out}}$ are set to be a multiple of $H$); the self-attention operation above is computed independently for the $H$ sets of query/key/value vectors. The results from each head (each of size $d_{\text{out}}/H$) are concatenated to produce the output $\boldsymbol{y}_t \in \mathbb{R}^{d_{\text{out}}}$.

One important distinction between the transformer and RNNs (Sec. 2.1) is their computational complexities. As we can see in Eq. 8, the size of key and value memory matrices linearly grows with the time step (i.e., sequence length)—unlike in the conventional RNN whose state has a constant size. This results in quadratic time complexity w.r.t. sequence length in the attention computation of Eq. 9—whereas complexity is linear in RNNs (i.e., compute is constant per time step). Consequently, practical self-attention requires predetermining a maximum sequence length, also called context window size, and any old events that fall outside the window are discarded. On the other hand, training of a transformer can be highly efficient. All the computations above are parallelizable over the sequence element by introducing the so-called "attention mask" inside the softmax as follows. By denoting an input sequence with $T$ elements as $\boldsymbol{X} = [\boldsymbol{x}_1, ..., \boldsymbol{x}_T] \in \mathbb{R}^{d_{\text{in}} \times T}$ ($\boldsymbol{X}_t = \boldsymbol{x}_t$ for all t) and the outputs as $\boldsymbol{Y} = [\boldsymbol{y}_1, ..., \boldsymbol{y}_T] \in \mathbb{R}^{d_{\text{out}} \times T}$ (and by analogously denoting queries, keys, and values for $T$ steps as $\boldsymbol{Q}, \boldsymbol{K} \in \mathbb{R}^{d_{\text{key}} \times T}$, and $\boldsymbol{V} \in \mathbb{R}^{d_{\text{out}} \times T}$, respectively), parallel computation performs:

$$\boldsymbol{Q} = \boldsymbol{W}^Q \boldsymbol{X} \ ; \ \boldsymbol{K} = \boldsymbol{W}^K \boldsymbol{X} \ ; \ \boldsymbol{V} = \boldsymbol{W}^V \boldsymbol{X} \tag{11}$$

$$\boldsymbol{Y} = \boldsymbol{V} \text{softmax}(\boldsymbol{M} \odot (\boldsymbol{K}^\top \boldsymbol{Q})) \tag{12}$$

where $\boldsymbol{M} \in \mathbb{R}^{T \times T}$ is the so-called attention mask. These equations are equivalent to the sequential Eqs. 7-9 by setting $\boldsymbol{M}$ to be the upper triangular matrix, i.e., $\boldsymbol{M}_{i,j} = 1$ if $i \leq j$ and $\boldsymbol{M}_{i,j} = -\infty$ otherwise; which explicitly sets certain attention weights to zero, as the causal model cannot access data from the future.

Practical implementations of transformers have been highly optimized. In particular, an efficient hardware-aware implementation is available (Dao, 2023), leveraging the "online softmax algorithm" (Rabe and Staats, 2021; Milakov and Gimelshein, 2018), which significantly reduces memory requirement (crucial for GPU

---

[2]It has been recognized that Eq. 9 can also be interpreted as a single-step iteration that minimizes a special energy function defining an Amari-Little-Hopfield network; we refer to Ramsauer et al. (2021) for further details.

efficiency) compared to the naive algorithm that explicitly stores $\boldsymbol{K}^\top \boldsymbol{Q} \in \mathbb{R}^{T \times T}$ in Eq. 12; we refer to Dao (2023) for further details.

# 3 Fast Weight Programmer Neural Networks

Here we present the concept of fast weight programming, and the resulting sequence models as well as their key properties.

## 3.1 Basic Instantiation: FWP with a purely additive update rule

Before introducing the general definition of fast weight programmers in the next Sec. 3.2. Here we first provide the most basic instantiation of FWPs: an "FWP with a purely additive outer product update rule" (Schmidhuber, 1992b) as an illustrative example; we refer to it as "vanilla FWP".

Like the conventional RNN and transformer, a vanilla FWP is a general-purpose sequence model. At every time step $t$, the model receives an input $\boldsymbol{x}_t \in \mathbb{R}^{d_{\text{in}}}$ and produces an output $\boldsymbol{y}_t \in \mathbb{R}^{d_{\text{out}}}$, while maintaining the so-called "fast weight" matrix $\boldsymbol{W}_t \in \mathbb{R}^{d_{\text{out}} \times d_{\text{key}}}$ as a short-term memory storage, as follows:

$$\boldsymbol{q}_t = \boldsymbol{W}^Q \boldsymbol{x}_t \; ; \; \boldsymbol{k}_t = \boldsymbol{W}^K \boldsymbol{x}_t \; ; \; \boldsymbol{v}_t = \boldsymbol{W}^V \boldsymbol{x}_t \tag{7}$$

$$\boldsymbol{W}_t = \boldsymbol{W}_{t-1} + \boldsymbol{v}_t \otimes \phi(\boldsymbol{k}_t) \tag{13}$$

$$\boldsymbol{y}_t = \boldsymbol{W}_t \phi(\boldsymbol{q}_t) \tag{14}$$

where Eq. 7 is the same as in the transformer (Sec. 2.3) with trainable parameters $\boldsymbol{W}^Q \in \mathbb{R}^{d_{\text{key}} \times d_{\text{in}}}$, $\boldsymbol{W}^K \in \mathbb{R}^{d_{\text{key}} \times d_{\text{in}}}$, and $\boldsymbol{W}^V \in \mathbb{R}^{d_{\text{out}} \times d_{\text{in}}}$; and as we'll discuss in Sec. 3.3, the connection between this model and the transformer does not end here. $\phi$ is an activation function we discuss later (while the activation on $\boldsymbol{v}$ is optional, in practice it is often also applied to $\boldsymbol{v}$). The "fast weight matrix" $\boldsymbol{W}_t \in \mathbb{R}^{d_{\text{out}} \times d_{\text{key}}}$ in Eq. 13 is initially set to 0, i.e., $\boldsymbol{W}_0 = 0$. See Figure 1b for illustration.

This model can be viewed as a system of two networks (Schmidhuber, 1992b) where one net—the slow net, corresponding to Eq. 7 (note that the three equations in Eq. 7 can be grouped into a single matrix multiplication by defining a "slow weight" matrix $\boldsymbol{W}^{\text{slow}} = [\boldsymbol{W}^Q, \boldsymbol{W}^K, \boldsymbol{W}^V] \in \mathbb{R}^{(2*d_{\text{key}} + d_{\text{out}}) \times d_{\text{key}}}$ by row concatenation)—learns to program or train another net, the fast net (Eq. 14) by generating its weight changes (Eq. 13). The fast weight change is defined through an update rule; here Eq. 13 takes the functional form of a Hebbian-like learning rule (Konorski, 1948; Hebb, 1949) whose update term is a simple outer product term.[3] The connection to SSMs (Sec. 2.2) is also noticeable: Eq. 13 is essentially a linear RNN with 2-dimensional state $\boldsymbol{W}_t$ (which could potentially be flattened and operationalized to be a one-dimensional vector state).

From a neuroscientific viewpoint, we may interpret this mechanism as rapid synaptic modulation (Panichello et al., 2024; Spaak and Wolff, 2025)—a property which is absent in the conventional RNNs (Sec. 2.1). Under this view, it might be natural not to consider the fast and slow net as representing the same type of "neural network" but rather, it may be more appropriate to consider the slow net as representing some "molecular network"—which is reminiscent of Denis Bray's view on ANNs (Bray, 1995; 2003; 2009), implementing molecular mechanisms that support learning and memory in the (fast) neural network. We provide further neurobiological discussion in Sec. 4.

From the memory system perspective, Eqs. 13-14 also correspond to an associative memory storing key/value pairs—also called correlation matrix memory (Kohonen, 1972) (see also Steinbuch and Piske (1963); Willshaw et al. (1969)), whose writing operation is an outer product of the key and value vectors (Eq. 13); and its reading/retrieval operation is a matrix-vector multiplication between the memory matrix and a query vector (Eq. 14). From a cognitive science view point, this model can also be seen as a learnable 2D version of the "tensor product representations" (outer product is the 2D tensor product) (Smolensky, 1990; Schlag and Schmidhuber, 2018; Schlag et al., 2019; 2021b)—an ANN model to bind two pieces of information (Greff et al., 2020).

---

[3]Indeed, a very similar model was more recently proposed (Limbacher and Legenstein, 2020), with a neuroscientific motivation to leverage Hebbian plasticity for sequence processing, and later extended to spiking neural networks (Limbacher et al., 2023); see also Najarro and Risi (2020).

## 3.2 Core Concept

As is examplified by the model in Sec. 3.1, a fast weight programmer (FWP) (Schmidhuber, 1992b; Irie et al., 2021) is defined as a neural network system in which one (sub)network, called slow net, generates modifications to weights of another (sub)network, called fast net, as a function of a sequence of input observations. By conceptualizing weights of a neural network as its program/software (Schmidhuber, 1990a), such a slow net which modifies the weights of a fast net is a *programmer*. The weights of the fast net are *fast*, because they can rapidly change as a response to observations received at every time step, while those of the slow net are *slow* because they are typically trained by some learning algorithm that updates slow weights on the sequence level (and they typically become fixed after training).[4]

In most of the existing and practical FWP models, both fast and slow nets are one-layer networks, and the weight update rule is some outer-product based one. However, the general concept of fast weight programming is more general and has no such a restriction (Irie et al., 2021); in principle, more complex (e.g., deeper) slow/fast nets or other update rules could be used. In fact, the core concept of FWPs is the idea of training a network to train (e.g., generate weights of) another network—an idea which has been rebranded as "hypernetworks" (Ha et al., 2017) in the modern deep learning literature. One common challenge though is to deal with the high-dimensionality of ANN weights, which are typically too large to be directly parameterized as outputs of an ANN (except for tiny networks (Gomez and Schmidhuber, 2005)). The use of outer product elegantly overcomes this challenge by generating two small vectors instead of one large matrix, and it's arguably more practical compared to other alternative methods that rely on weight compression (Irie and Schmidhuber, 2021) or sparsity (Munkhdalai, 2020).

Another issue is that naively applying the BPTT learning algorithm to FWP would require storing all the intermediate weight matrices for each time step for backpropagation, which would yield memory requirements that can easily exceed the amount of memory available on GPUs. Therefore, practical FWP model designs also require compute-efficient recomputability of fast weights (e.g., through reversibility of the update rule; we refer to the corresponding discussions in prior work (Schlag et al., 2021a; Irie and Schmidhuber, 2021)) or chunk-wise processing (see Box 1).

As a historical note, McCulloch and Pitts (1943) also discussed recurrence and dynamic synapses (which they called 'circle' and 'alterable synapses', respectively) in their seminal paper on ANNs. Their proposal was to replace dynamic synapses by recurrence which can potentially simulate the effect of dynamic synapses (see their informal "theorem 7"). However, fixed weights of ANNs were later criticized as a limitation from both the theoretical neuroscience and machine learning standpoints (von der Malsburg, 1981; Feldman, 1982; McClelland, 1985), and the possibility to introduce fast synaptic modulations was investigated. In particular, von der Malsburg (1981) and Hinton and Plaut (1987) proposed networks whose effective weights are defined as a multiplicative (von der Malsburg, 1981) or additive (Hinton and Plaut, 1987) superposition of fast and slow changing weights. However, none of these early works has proposed a mechanism to learn the dynamics of synapses (e.g., Hinton and Plaut (1987) merely used two different learning rates for the fast and slow weights, while jointly training both sets of weights through the same gradient descent learning algorithm). It was only in the early 1990s that end-to-end differentiable and learnable synaptic modulation dynamics above (originally called "fast weight controllers"), featuring the "programming" part—the 'P' in FWP—was proposed (Schmidhuber, 1991; 1992b), as an alternative to the conventional recurrence (Sec. 2.1) and was also computationally motivated by the idea of reducing the ratio of trainable parameters to temporally changing variables in sequence processing networks (Schmidhuber, 1993b). The FWP concept has seen a recent revival (Schmidhuber, AI Blog, 2021; Irie and Schmidhuber, 2022) mainly due to its formal connection to the transformer and its potential to overcome certain limitations of transformers (as we'll see in the next sections).

---

[4]Note that this fast vs. slow distinction may not hold in some edge cases where some fully online learning algorithm, such as real-time recurrent learning with an update frequency of one time step, is used to learn the slow weights (in which case the slow weights are also updated at every time step)—even though no such learning algorithm is common in practice; see, e.g., Irie et al. (2024). Nevertheless, this terminology is conceptually appropriate and illustrative of the characteristic timescale difference underlying FWPs.

### 3.3 Formal Connection to Transformers

Here we present the formal connection between the vanilla FWP (Sec. 3.1) and the transformer (Sec. 2.3).[5] We discuss this relation in two didactic steps by looking into (1) a transformer without softmax, and (2) a transformer with an alternative (but still normalized) attention-score function.

**Transformer without softmax.** First, we examine the consequence of simply removing the softmax in the self-attention of Eq. 9, which yields:

$$\boldsymbol{y}_t = \boldsymbol{V}_t(\boldsymbol{K}_t^\top \boldsymbol{q}_t) = (\boldsymbol{V}_t \boldsymbol{K}_t^\top)\boldsymbol{q}_t \tag{15}$$

The removal of softmax opens up the possibility to reorganize the computation by first multiplying $\boldsymbol{V}_t \boldsymbol{K}_t^\top$ before multiplying it with the query $\boldsymbol{q}_t$. By denoting this key-value product as $\boldsymbol{W}_t = \boldsymbol{V}_t \boldsymbol{K}_t^\top \in \mathbb{R}^{d_{\text{out}} \times d_{\text{key}}}$, we can further express it in terms of the column vectors in each of the key and value matrices (recall their definition of Eq. 8) as in the following Eq. 16:

$$\boldsymbol{W}_t = \boldsymbol{V}_t \boldsymbol{K}_t^\top = \sum_{\tau=1}^{t} \boldsymbol{v}_\tau \otimes \boldsymbol{k}_\tau \tag{16}$$

$$= \boldsymbol{W}_{t-1} + \boldsymbol{v}_t \otimes \boldsymbol{k}_t \tag{17}$$

Further isolating the last term in the sum of Eq. 16 yields the above Eq. 17 which expresses a recurrent formula for $\boldsymbol{W}_t$. Overall, the sequential dynamic of the transformer without softmax can be rewritten as:

$$\boldsymbol{q}_t = \boldsymbol{W}^Q \boldsymbol{x}_t \; ; \; \boldsymbol{k}_t = \boldsymbol{W}^K \boldsymbol{x}_t \; ; \; \boldsymbol{v}_t = \boldsymbol{W}^V \boldsymbol{x}_t \tag{7}$$

$$\boldsymbol{W}_t = \boldsymbol{W}_{t-1} + \boldsymbol{v}_t \otimes \boldsymbol{k}_t \tag{17}$$

$$\boldsymbol{y}_t = \boldsymbol{W}_t \boldsymbol{q}_t \tag{18}$$

which we can recognize as being identical to the vanilla FWP in Sec. 3.1, up to the missing activation function applied to $\boldsymbol{q}_t$ and $\boldsymbol{k}_t$. This means that, the exact input/output mapping of a transformer without softmax can be equivalently expressed by an FWP.

This equivalence result may be intriguing at first, because even without the softmax, the transformer model stores a key-value memory storage that grows with the sequence length (potentially to infinity), while the FWP has a fixed-size memory storage in the fast weight matrix (compare Figure 1b with Figure 1c). This highlights the role of softmax as a powerful retrieval function enabling sharp discrimination between a large set of memory elements; without such a discriminative retrieval function, a key-value memory system with even an infinitely growing memory size is merely as powerful as an FWP system with a fixed memory size.

**Transformer with linearized attention (Linear transformer).** While the derivation above based on the simple removal of softmax is straightforward and captures the core matrix-algebra manipulation underlying the equivalence between the transformer and the vanilla FWP, we can also revisit the removal of the softmax, instead replacing the softmax-normalized attention score computation softmax($\boldsymbol{K}_t^\top \boldsymbol{q}_t$) in Eq. 9 (the softmax kernel) by another kernel function, which computes the normalized attention scores (for $\tau$ from 1 to $t$) as:

$$\boldsymbol{\alpha}'_{t,\tau} = \frac{\phi(\boldsymbol{k}_\tau)^\top \phi(\boldsymbol{q}_t)}{\sum_{\tau'=1}^{t} \phi(\boldsymbol{k}_{\tau'})^\top \phi(\boldsymbol{q}_t)} \tag{19}$$

for an arbitrary function $\phi$ with a positive co-domain. Compared to the case above where we simply removed the softmax, we have extra $\phi$ applied to the keys and the query, and the denominator that normalizes the attention score. Despite these differences, we can similarly reorganize the computations in the corresponding

---

[5]Hinton (2022) asks: "For sequential data, is it possible to use fast weights to mimic a simplified transformer?" The answer is yes, as shown in Schlag et al. (2021a), which we review here.

self-attention computation as follows:

$$\boldsymbol{y}_t = \sum_{\tau=1}^{t} \boldsymbol{\alpha}'_{t,\tau} \boldsymbol{v}_\tau = \frac{\sum_{\tau=1}^{t} \boldsymbol{v}_\tau \phi(\boldsymbol{k}_\tau)^\top \phi(\boldsymbol{q}_t)}{\sum_{\tau'=1}^{t} \phi(\boldsymbol{k}_{\tau'})^\top \phi(\boldsymbol{q}_t)} = \frac{\left(\sum_{\tau=1}^{t} \boldsymbol{v}_\tau \otimes \phi(\boldsymbol{k}_\tau)\right) \phi(\boldsymbol{q}_t)}{\left(\sum_{\tau'=1}^{t} \phi(\boldsymbol{k}_{\tau'})\right)^\top \phi(\boldsymbol{q}_t)} \tag{20}$$

$$= \frac{1}{\boldsymbol{z}_t^\top \phi(\boldsymbol{q}_t)} \boldsymbol{W}_t \phi(\boldsymbol{q}_t) \tag{21}$$

where, in the numerator, $\boldsymbol{W}_t$ is defined as $\boldsymbol{W}_t = \sum_{\tau=1}^{t} \boldsymbol{v}_\tau \otimes \phi(\boldsymbol{k}_\tau) \in \mathbb{R}^{d_{\text{out}} \times d_{\text{key}}}$ whose recurrent update function can be derived similarly to Eqs. 16-17, and $\boldsymbol{z}_t \in \mathbb{R}^{d_{\text{key}}}$ in the denominator of Eq. 21 has the following recurrent update equation with $\boldsymbol{z}_0 = 0$:

$$\boldsymbol{z}_t = \sum_{\tau'=1}^{t} \phi(\boldsymbol{k}_{\tau'}) = \boldsymbol{z}_{t-1} + \phi(\boldsymbol{k}_t) \tag{22}$$

Overall, the sequential dynamics of a transformer model based on an alternative normalized attention function defined by Eq. 19 can be rewritten as:

$$\boldsymbol{q}_t = \boldsymbol{W}^Q \boldsymbol{x}_t \; ; \; \boldsymbol{k}_t = \boldsymbol{W}^K \boldsymbol{x}_t \; ; \; \boldsymbol{v}_t = \boldsymbol{W}^V \boldsymbol{x}_t \tag{7}$$

$$\boldsymbol{W}_t = \boldsymbol{W}_{t-1} + \boldsymbol{v}_t \otimes \phi(\boldsymbol{k}_t) \tag{13}$$

$$\boldsymbol{z}_t = \boldsymbol{z}_{t-1} + \phi(\boldsymbol{k}_t) \tag{22}$$

$$\boldsymbol{y}_t = \frac{1}{\boldsymbol{z}_t^\top \phi(\boldsymbol{q}_t)} \boldsymbol{W}_t \phi(\boldsymbol{q}_t) \tag{21}$$

This is the "recurrent form" of the so-called "linear transformer" (Katharopoulos et al., 2020). This system is identical to the FWP of Sec. 3.1 up to the normalizing denominator $\boldsymbol{z}_t^\top \phi(\boldsymbol{q}_t) \in \mathbb{R}$ in Eq. 21 and tracking of the extra time-varying variable $\boldsymbol{z}_t$ (Eq. 22). From this view, the vanilla FWP is essentially an "unnormalized linear transformer" (ULTRA). In fact, recent work extending linear transformer models (discussed in Sec. 3.4) has shown that such normalization is unnecessary in practice (Schlag et al., 2021a; Sun et al., 2023; Yang et al., 2024a;b).

While "linear" in the name "linear transformer" could highlight how the linearized attention function of Eq. 19 allows for its computation to be reorganized unlike the softmax attention, it primarily refers to its time complexity. Unlike the quadratic complexity of the softmax attention (Sec. 2.3), the computation per step is constant w.r.t. the time step/sequence length in this model; the resulting time complexity is linear w.r.t. sequence length like with RNNs. This is an example of efficient sequence models as its training is parallelizable using the "attention form"—parallel computation analogous to that of softmax attention (Eq. 12) can be derived, while its inference is linear-time complexity by using the recurrent form above. Naturally, such a computational advantage comes with a cost: performance of the linear transformer largely lags behind that of the quadratic transformer in practice (Katharopoulos et al., 2020; Schlag et al., 2021a). However, as we'll see in the next Sec. 3.4, more effective but still efficient models can be derived by extending the vanilla FWP model.

As a side note, while the original linear transformer by Katharopoulos et al. (2020) simply used $\phi(\boldsymbol{x}) = \text{ELU}(\boldsymbol{x}) + 1$ (where ELU denotes "exponential linear unit" (Clevert et al., 2016)), both Choromanski et al. (2021) and Peng et al. (2021) proposed a linear transformer that uses random feature kernels (i.e., $\phi$ is not just a judiciously chosen activation function but also involves up-projection using randomly sampled features) which are formal approximations of the softmax in theory. However, such approximation methods (which hold with infinite many random features) do not perform well in practical scenarios.

As a historical note, while the derivation of the recurrent form of the linear transformer from its attention form was provided by Katharopoulos et al. (2020) (2020), the same mathematical derivation can also be found in Ba et al. (2016a) (2016) which connected a special instantiation of recurrent FWPs (Schmidhuber, 1993b) (in which a recurrent hidden state is used as both keys and values) to a form of attention. More broadly speaking, the mathematical derivation relating the vanilla FWP to unnormalized attention is the

Table 1: A few variations of Fast Weight Programmers with the corresponding update rules and underlying local (minimized) objective functions. $\boldsymbol{W}_t$, $\boldsymbol{W}_{t-1}$, and $\boldsymbol{W}$ are matrices; $\boldsymbol{v}_t$ and $\boldsymbol{k}_t$ are vectors, $\boldsymbol{a}_t$ is a vector with elements in $(0,1)$; $\mathbf{1}$ denotes a vector whose elements are all one; $\lambda$ and $\lambda_t$ are scalars in $(0,1)$; $\eta_t$ is a non-negative scalar serving as a learning rate in the update rule (the update rules that do not involve $\eta_t$ use a learning rate of 1). $\otimes$ and $\odot$ denote outer product and element-wise/Hadamard product, respectively. Derivations can be found in Appendix A.

| Model | State Update Rule | Local Loss $\mathcal{L}_t(\boldsymbol{W})$ |
|---|---|---|
| Vanilla FWP | $\boldsymbol{W}_t = \boldsymbol{W}_{t-1} + \boldsymbol{v}_t \otimes \phi(\boldsymbol{k}_t)$ | $-\boldsymbol{v}_t^\top \boldsymbol{W} \phi(\boldsymbol{k}_t)$ |
| *Use Classic Rule* | | |
| DeltaNet (Schlag et al., 2021a) | $\boldsymbol{W}_t = \boldsymbol{W}_{t-1} + \eta_t(\boldsymbol{v}_t - \boldsymbol{W}_{t-1}\phi(\boldsymbol{k}_t)) \otimes \phi(\boldsymbol{k}_t)$ | $\frac{1}{2}\|\boldsymbol{v}_t - \boldsymbol{W}\phi(\boldsymbol{k}_t)\|_2^2$ |
| OjaNet (Irie et al., 2022b) | $\boldsymbol{W}_t = \boldsymbol{W}_{t-1} + \eta_t \boldsymbol{v}_t \otimes (\phi(\boldsymbol{k}_t) - \boldsymbol{W}_{t-1}^\top \boldsymbol{v}_t)$ | $-\boldsymbol{v}_t^\top \boldsymbol{W}\phi(\boldsymbol{k}_t) + \frac{1}{2}\|\boldsymbol{W}^\top \boldsymbol{v}_t\|_2^2$ |
| *Introduce Decaying* | | |
| RetNet Sun et al. (2023) | $\boldsymbol{W}_t = \lambda \boldsymbol{W}_{t-1} + \boldsymbol{v}_t \otimes \phi(\boldsymbol{k}_t)$ | $-\boldsymbol{v}_t^\top \boldsymbol{W}\phi(\boldsymbol{k}_t) + \frac{1-\lambda}{2}\|\boldsymbol{W}\|_F^2$ |
| Mamba2 (Dao and Gu, 2024) | $\boldsymbol{W}_t = \lambda_t \boldsymbol{W}_{t-1} + \boldsymbol{v}_t \otimes \phi(\boldsymbol{k}_t)$ | $-\boldsymbol{v}_t^\top \boldsymbol{W}\phi(\boldsymbol{k}_t) + \frac{1-\lambda_t}{2}\|\boldsymbol{W}\|_F^2$ |
| Gated RFA (Peng et al., 2021) | $\boldsymbol{W}_t = \lambda_t \boldsymbol{W}_{t-1} + (1-\lambda_t)\boldsymbol{v}_t \otimes \phi(\boldsymbol{k}_t)$ | $-(1-\lambda_t)\boldsymbol{v}_t^\top \boldsymbol{W}\phi(\boldsymbol{k}_t) + \frac{1-\lambda_t}{2}\|\boldsymbol{W}\|_F^2$ |
| mLSTM in xLSTM (Beck et al., 2024) | $\boldsymbol{W}_t = \lambda_t \boldsymbol{W}_{t-1} + \eta_t \boldsymbol{v}_t \otimes \phi(\boldsymbol{k}_t)$ | $-\eta_t \boldsymbol{v}_t^\top \boldsymbol{W}\phi(\boldsymbol{k}_t) + \frac{1-\lambda_t}{2}\|\boldsymbol{W}\|_F^2$ |
| GLA (Yang et al., 2024a) | $\boldsymbol{W}_t = (\boldsymbol{a}_t \otimes \mathbf{1}) \odot \boldsymbol{W}_{t-1} + \boldsymbol{v}_t \otimes \phi(\boldsymbol{k}_t)$ | $-\boldsymbol{v}_t^\top \boldsymbol{W}\phi(\boldsymbol{k}_t) + \frac{1}{2}\|((\sqrt{1-\boldsymbol{a}_t}) \otimes \mathbf{1}) \odot \boldsymbol{W}\|_F^2$ |
| *Combine Methods* | | |
| Gated DeltaNet (Yang et al., 2025) | $\boldsymbol{W}_t = \lambda_t \boldsymbol{W}_{t-1} + \eta_t(\boldsymbol{v}_t - \boldsymbol{W}_{t-1}\phi(\boldsymbol{k}_t)) \otimes \phi(\boldsymbol{k}_t)$ | $\frac{1}{2}\|\boldsymbol{v}_t - \boldsymbol{W}\phi(\boldsymbol{k}_t)\|_2^2 + \frac{1-\lambda_t}{2\eta_t}\|\boldsymbol{W}\|_F^2$ |

same as the classic derivation in ML that derives the duality between the perception and its dual form, kernel machines by Aizerman et al. (1964) (1964); based on this parallel, the vanilla FWP computation is the *primal form*, and (unnormalized) attention is its *dual form* (Irie et al., 2022a).

A practical implication of this equivalence is that FWPs are typically used as a drop-in replacement to the self-attention operation in the transformer architectures, while preserving other transformer components, including two-layer feedforward blocks, layer normalization, residual connections, as well as the use of multiple heads (Vaswani et al., 2017).

### 3.4 Going beyond the vanilla FWP: exploring fast weight update rules

A core characteristics of FWPs—the use of an update rule that has a form of a learning rule in the forward dynamics of the system to train a subnetwork on the fly—naturally motivates us to explore and improve the update rule used in Eq. 13.

For example, one idea is to replace the purely Hebbian-like learning rule of Eq. 13 by the error-correcting delta-rule (Widrow and Hoff, 1960). This yields the following model, called "DeltaNet" (Schlag et al., 2021a):

$$\boldsymbol{q}_t = \boldsymbol{W}^Q \boldsymbol{x}_t \; ; \; \boldsymbol{k}_t = \boldsymbol{W}^K \boldsymbol{x}_t \; ; \; \boldsymbol{v}_t = \boldsymbol{W}^V \boldsymbol{x}_t \; ; \; \beta_t = \boldsymbol{w}^{b\top} \boldsymbol{x}_t \tag{23}$$

$$\boldsymbol{W}_t = \boldsymbol{W}_{t-1} + \psi(\beta_t)(\boldsymbol{v}_t - \boldsymbol{W}_{t-1}\phi(\boldsymbol{k}_t)) \otimes \phi(\boldsymbol{k}_t) \tag{24}$$

$$\boldsymbol{y}_t = \boldsymbol{W}_t \phi(\boldsymbol{q}_t) \tag{14}$$

where in addition to the query/key/value generated by the slow net in Eq. 23, an extra trainable parameter vector $\boldsymbol{w}^b \in \mathbb{R}^{d_{\text{in}}}$ is introduced to generate a scalar variable $\beta_t \in \mathbb{R}$, which will serve as a dynamic learning rate (in the multi-head case, different learning rates are generated for each head); in Eq. 24, $\psi$ is typically set to 2 times the sigmoid function (the factor 2 is crucial to introduce negative eigenvalues in the state transition matrix enabling improved expressivity (Grazzi et al., 2025); we discuss in Sec. 3.6).

Eq. 24 corresponds to a rank-one update of the fast weight matrix, from $\boldsymbol{W}_{t-1}$ to $\boldsymbol{W}_t$, through the delta learning rule (Widrow and Hoff, 1960), where the slow net-generated variables, $\boldsymbol{v}_t$, $\phi(\boldsymbol{k}_t)$, and $\psi(\beta_t)$, play the role of *target*, *input*, and *learning rate* of the delta rule, respectively. See Box 3 for further comments on the delta rule.

From the memory system perspective (Schlag et al., 2021a), this update rule can also be interpreted as follows: instead of naively adding the new key-value association $(\boldsymbol{k}_t, \boldsymbol{v}_t)$ to be stored in memory (as is the case for the purely additive rule of Eq. 13), we first check the old value that is currently associated to the new key $\boldsymbol{k}_t$ by querying the current memory matrix $\boldsymbol{W}_{t-1}\phi(\boldsymbol{k}_t)$; which we remove from the memory, while adding the new value $\boldsymbol{v}_t$. The effective residual value vector to be added to the memory is their "delta", i.e., $\boldsymbol{v}_t - \boldsymbol{W}_{t-1}\phi(\boldsymbol{k}_t)$.

In practice, DeltaNet has been shown to consistently outperform the vanilla FWP with the purely additive update rule (Sec. 3.1) across many tasks including language modeling (Schlag et al., 2021a; Irie et al., 2021; Yang et al., 2024b), reinforcement learning in game environments (Irie et al., 2021), time series classification (Irie et al., 2022b), and image generation (Irie and Schmidhuber, 2023). One natural question is whether DeltaNet is still efficient, i.e., whether its training is parallelizable, and the answer is yes; Yang et al. (2024b) have derived a parallel training algorithm for DeltaNet.

More broadly, many recently proposed efficient sequence models—such as Gated Linear Attention (GLA) (Yang et al., 2024a), Mamba2 (Dao and Gu, 2024), RetNet (Sun et al., 2023), mLSTM in xLSTM (Beck et al., 2024), and Gated DeltaNet (Yang et al., 2025)—can also be expressed as an FWP with a specific state/weight update rule; the corresponding summary is shown in Table 1. This FWP view facilitates relating and comparing these models—relations which may be nebulous solely from their names. For example, we can categorize that many of these models simply introduce a decay factor on the weight/state and differ from each other in the type of decay used: RetNet uses a constant scalar decay, whereas Mamba2 uses a context/time-dependent scalar (produced as a function of the input; similarly to the dynamic learning rate of DeltaNet in Eq. 23), while GLA dynamically produces different decay rates for each row of the fast weight matrix. Oja's rule (Oja, 1982) is also a natural extension to the naive Hebbian rule; however, OjaNet was reported to empirically underperform DeltaNet on certain sequence processing applications (Irie et al., 2022b); which may be an intuitive result as Oja's rule performs principal component analysis (Oja, 1982), while the delta rule is for error correction. Certain other rules can be naturally derived as extensions of the delta rule (Yang et al., 2025; Peng et al., 2025). Further discussions of local objectives underlying different update rules are provided in the next section and in Table 1.

As a side note, some of the early FWP-like models whose development predates 2020 (Schlag and Schmidhuber, 2017; Munkhdalai and Yu, 2017; Munkhdalai and Trischler, 2018; Miconi et al., 2018; 2019; Keller et al., 2018; Munkhdalai et al., 2019) (at the time when sequence model development was much less dominated by the training efficiency; we remind that the GPT-2 (Radford et al., 2019) and GPT-3 (Brown et al., 2020) language models were introduced in 2019 and 2020, respectively), are somewhat harder to fit in this table, as they tended to use fast weights within the LSTM architecture; but we can find the same core idea: replacing certain weight matrices in the LSTM by fast weights modified over time through an update rule.

**Practical considerations.** To determining the to-go model for a specific task, our current recommendation is to try both DeltaNet and Gated DeltaNet variants (Table 1): while Gated DeltaNet has been reported to outperform DeltaNet on language modeling tasks, consistency of this advantage in other tasks has not been confirmed yet (e.g., for reinforcement learning in certain game environments, we found weight/memory decay of the gated variant to hurt; unpublished work). Our general recommendation is to avoid relying solely on existing language modeling results when applying FWPs as general-purpose sequence models to other tasks.

As for practical considerations, the choice of $\phi$ (applied to key and query vectors) has a direct impact on both good performance and stability (especially when the delta rule is used (Schlag et al., 2021a)). Our current recommendation is to set $\phi$ to be the element-wise sigmoid linear unit (SiLU $= \boldsymbol{x} \odot \text{sigmoid}(\boldsymbol{x})$) followed by the $L_2$ normalization as proposed by Yang et al. (2024b). Further discussion on the practical training algorithm can be found in Box 1.

Efficient implementations for most models listed in Table 1 are openly available on the actively maintained "flash-linear-attention" repository (Yang and Zhang, 2024); using these models are typically as easy as using an LSTM in PyTorch, and could be a good starting point for any other FWP model development.

---

**Box 1: Chunk-wise Parallel Training Algorithm for FWPs**

In practice, the FWP models discussed here are trained using a so-called "chunk-wise parallel training" algorithm, which is a hybrid approach leveraging both the recurrent and attention form of FWPs (Hua et al., 2022; Sun et al., 2023; Yang et al., 2024a). While the exact algorithm is derived for each FWP model, the main idea is to divide a training sequence into small chunks and causally

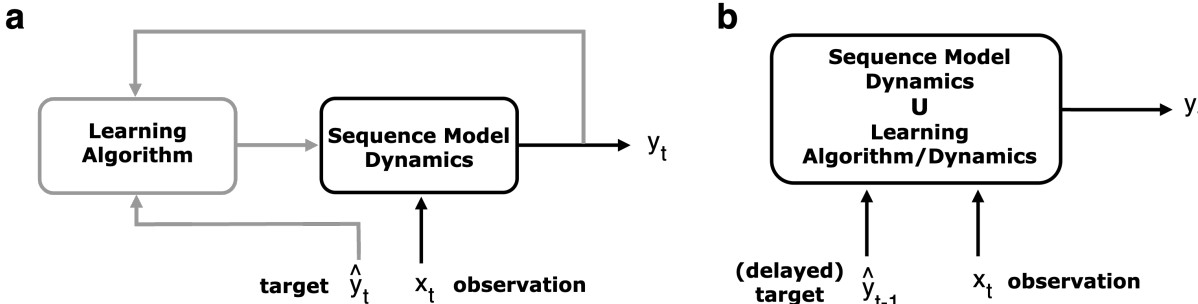

Figure 2: An Illustration constrasting **a:** a conventional view on sequence model with a learning algorithm, and **b:** a metalearned (or in-context learning) system that embeds learning algorithms/dynamics within its sequential dynamics. In **a**, the sequence model only observes an input, and produces an output (**black**), while the learning algorithm receives the expected target and the model output, and takes care of adjusting the parameters of the sequence model to improve upon the given task (**gray**). In contrast, in **b**, the system itself observes the input and the (delayed) expected target, and self-improvement on the task, i.e., learning, is part of its sequential dynamics.

process one chunk after another; the intra-chunk computation leverages the parallel computation, while the inter-chunk contributions are computed using the recurrent form.

Here we illustrate the main idea by focusing on the algorithm for the vanilla FWP with $\phi$ set to identity. Let $S$ and $\mathbf{n}$ denote positive integers. We denote all the model outputs in the $\mathbf{n}$-th chunk of size $S$ as $\mathbf{Y_n} \in \mathbb{R}^{d_{\text{out}} \times S}$, which can be computed as:

$$\mathbf{Y_n} = \mathbf{W_n Q_n} + \mathbf{V_n}(\mathbf{K_n^\top Q_n} \odot \mathbf{M}) \;\; ; \;\; \mathbf{W_{n+1}} = \mathbf{W_n} + \mathbf{V_n K_n^\top} \tag{25}$$

where $\mathbf{Q_n}$, $\mathbf{K_n} \in \mathbb{R}^{d_{\text{key}} \times S}$ and $\mathbf{V_n} \in \mathbb{R}^{d_{\text{out}} \times S}$ denote matrices containing all the query, key, value vectors for chunk $\mathbf{n}$, respectively, and $\mathbf{W_n} \in \mathbb{R}^{d_{\text{out}} \times d_{\text{key}}}$ is the state of the fast weight matrix after observing all the inputs from the beginning of the sequence up to chunk $\mathbf{n}$ (exclusive), with $\mathbf{W_0} = 0$, and $\mathbf{M} \in \mathbb{R}^{S \times S}$ is the causal mask. In Eq. 25, the first and second terms of the left equation correspond to the inter- and intra-chunk computations, respectively, while the right equation is the chunk-level fast weight update.

Analogous algorithms can be rather straightforwardly derived for all the FWPs based on weight decays (see Table 1). While non trivial, an algorithm for DeltaNet can also be derived (Yang et al., 2024b) and scales well in practice. Actual implementations for different models can be found, e.g., in the open-source code available at the flash-linear-attention repository (Yang and Zhang, 2024).

### 3.5 Local online learning, metalearning, and conception of in-context learning

The concept and structure of FWPs also offer unique insights into the idea of local online learning and metalearning, with implications for research on learning mechanisms compatible with biological constraints.

**Local online learning.** The structure of FWPs captures the fundamental idea of expressing the learning dynamics, i.e., the process of "training a network", within the model's sequential dynamics (Cotter and Conwell, 1990; 1991; Younger et al., 1999; 2001; Hochreiter et al., 2001b)—a slow net learns to "train" a fast net as a part of sequence processing. This view is further reinforced in the case of DeltaNet in which the classic delta rule—conventionally used in the "backward pass", i.e., in the process of learning the (slow) weights of an ML system—is used in the "forward pass" to perform online updates of the fast weights based

on the variables (input/target/learning rate) produced by the slow net on the fly; essentially performing a local online training of the fast net.

This local optimization aspect becomes even more prominent by explicitly writing down the local objective function that is optimized by the corresponding update rule. For example, the classic delta rule (Eq. 24) corresponds to the derivative (w.r.t. the fast net weights $\boldsymbol{W}_t$) of the squared loss $||\boldsymbol{v}_t - \boldsymbol{W}_t\phi(\boldsymbol{k}_t)||_2^2$ between the "target" $\boldsymbol{v}_t$ and the output $\boldsymbol{W}_t\phi(\boldsymbol{k}_t)$ that the fast net would produce if $\phi(\boldsymbol{k}_t)$ were fed to its input (which is consistent with the idea of binding $\phi(\boldsymbol{k}_t)$ to $\boldsymbol{v}_t$ by storing the corresponding key/value pair in the memory matrix $\boldsymbol{W}_t$); see Box 3 for further details. More generally, the update rules used in typical FWP models have a corresponding local objective function, as summarized in the last column of Table 1. For example, using a state/weight decay in the update rule corresponds to introducing the $L_2$ regularization on the fast weight matrix $||\boldsymbol{W}||_F^2$ in the local objective function (disregarding constant factors), where the regularization strengths are the weight decay factors.

As a side note, such an idea of an optimized model (the slow net itself is trained/optimized for an external objective function, e.g., by gradient descent) that internally optimizes a certain objective function is often called "mesa-optimization" (Hubinger et al., 2019) and the corresponding hidden objective is called "mesa-objective". A natural extension of such a view on FWPs has given rise to another class of FWP models, in which, instead of defining a single-step update rule, a local objective function is directly defined, and the model output is produced by finding the corresponding optimum by using an explicit optimizer. For further details, we refer to concrete examples of this model family, such as MesaNet (von Oswald et al., 2025; 2023b) and Titan (Behrouz et al., 2024; 2025a) (see also Behrouz et al. (2025b)), as well as the related concepts of "test-time training/regression" (Sun et al., 2025; Wang et al., 2025).

**Metalearning.**   The FWP concept of training a model to train another model (or itself) is also the essence of *metalearning* in ML (Schmidhuber, 1987; Chalmers, 1990; Bengio et al., 1991; Hochreiter et al., 2001b). While any general-purpose sequence models (including any models discussed in this Primer) can be potentially trained to become an online learner through metalearning (as we explain below), the structure of FWPs provides an intuitive conception: the slow net implements a learning algorithm for the fast net. Remarkably, von Oswald et al. (2023a) have effectively derived a specific slow weight configuration for a vanilla FWP to implement the gradient descent learning algorithm for regression problems in its forward dynamics (see Box 2 for further details).

One crucial ingredient for metalearning we have not discussed yet is the "error feedback". For any sequence model to become an online learner (capable of effectively learning new tasks through observations), error feedback needs to be provided to the model, in addition to the input observation. There are two common ways to do so. One way (which we call the "delayed-feedback" setting following Irie et al. (2022c)) is to feed the ground truth target from the previous time step as an additional input to the model (i.e., with a one time-step delay); in this case, the model continually receives an input $\boldsymbol{x}_t$ and a delayed feedback $\hat{\boldsymbol{y}}_{t-1}$ and predicts $\boldsymbol{y}_t$ at every time step (Hochreiter et al., 2001b; Santoro et al., 2016). Alternatively, in the "synchronous-feedback" setting (Mishra et al., 2018), we feed both an input $\boldsymbol{x}_t$ and the corresponding target $\boldsymbol{y}_t$ to the model at every time step as demonstrations of the task; and for an input on which we want the model to make a prediction, no target is provided; instead, we replace it with special values that indicate it is not a demonstration and that a prediction is being requested.

In both cases, such formulations turn the problem of learning itself into a sequence learning problem; by (meta-)training a sequence model on many such example sequences, each representing a different task (i.e., a different learning experience), we can obtain an online learner capable of learning a new task by observing some task demonstrations (i.e., pairs of an observation and the expected target from the task).

While such an online learning capabilities is often called *in-context learning* (Brown et al., 2020; Garg et al., 2022; Raventós et al., 2023) today, and it is often (misleadingly) described as a somewhat magical capability of transformer-based large language models (LLMs); by using some metalearning process like the one described above, we can meta-train any sequence model to become an online learner involving modalities beyond languages. For example, the seminal work by Hochreiter et al. (2001b) trained an LSTM to perform in-context regression—demonstrating the "fixed-weight learning" concept advocated by Cotter, Conwel, and Younger (Cotter and Conwell, 1990; 1991; Younger et al., 1999) in the early 1990s, while Santoro et al. (2016) and

Mishra et al. (2018) performed in-context image classification—all predating the term in-context learning. There are numerous such examples across tasks, modalities, and model architectures (Bosc, 2015; Santoro et al., 2016; Duan et al., 2016; Wang et al., 2017; Munkhdalai and Yu, 2017; Munkhdalai and Trischler, 2018; Mishra et al., 2018; Miconi et al., 2018; 2019; Munkhdalai et al., 2019; Sandler et al., 2021; Kirsch and Schmidhuber, 2021; Huisman et al., 2023), and they are not specific to the transformer architecture or the language modality.

From the metalearning perspective, it is not surprising that LLMs are capable of in-context learning, given that the task of auto-regressive next-token prediction—underlying language modeling—precisely follows the form required by metalearning: prediction with error feedback (the delayed-feedback version above); and the internet-scale text could provide data necessary for such meta-training (Irie and Lake, 2025).

---

**Box 2: Slow Weight Configuration Implementing Gradient Descent**

Here we review how von Oswald et al. (2023a) constructed a slow weight configuration that implements a gradient descent learning algorithm in the forward pass of the vanilla FWP (Sec. 3.1) for linear regression problems.

We consider a regression task with input and output dimensions $d_{\mathrm{x}}$ and $d_{\mathrm{y}}$, respectively, and the corresponding data set $(z_t, f(z_t))$ for $z_t \in \mathbb{R}^{d_{\mathrm{x}}}$ and $f(z_t) \in \mathbb{R}^{d_{\mathrm{y}}}$ for $t$ from 1 to $T$ with an unknown function $f$.

Let's first describe what the gradient descent algorithm would do for linear regression. If we had a linear model with a weight matrix $W_0 \in \mathbb{R}^{d_{\mathrm{y}} \times d_{\mathrm{x}}}$, and performed one step of gradient descent on the loss $\frac{1}{2} \sum_{t=1}^{T} \|f(z_t) - W_0 z_t\|_2^2$, the resulting weight matrix would be $W_0 + \Delta W_T$ with $\Delta W_T = \sum_{t=1}^{T} (f(z_t) - W_0 z_t) \otimes z_t$ (here we use a learning rate of 1 but the construction can be easily extended to the case with an arbitrary learning rate). Given a new input $z^\star \in \mathbb{R}^{d_{\mathrm{x}}}$, prediction of the linear model with the updated weight matrix would be $(W_0 + \Delta W_T) z^\star$, which can be further expressed as $(W_0 + \Delta W_T) z^\star \approx \Delta W_T z^\star$ for small initialization $W_0$.

The goal here is to construct a weight configuration of $W^Q$, $W^K$, $W^V$ in the vanilla FWP that can reproduce the algorithm above, by simply following the FWP equations of Sec. 3.1. This construction uses one-layer single-head vanilla FWP model with $\phi$ function set to identity. The feedback scheme is synchronous, that is, we feed to the model a sequence of demonstration vectors $x_t = [z_t, f(z_t)] \in \mathbb{R}^{d_{\mathrm{x}}+d_{\mathrm{y}}}$, each is a concatenation of an observation $z_t \in \mathbb{R}^{d_{\mathrm{x}}}$ and the ground truth target $f(z_t) \in \mathbb{R}^{d_{\mathrm{y}}}$.

The proposed weight configuration is the following:

$$W^Q = W^K = \left( \begin{array}{c|c} I_{d_{\mathrm{x}}} & 0_{d_{\mathrm{x}} \times d_{\mathrm{y}}} \\ \hline 0_{d_{\mathrm{y}} \times d_{\mathrm{x}}} & 0_{d_{\mathrm{y}} \times d_{\mathrm{y}}} \end{array} \right) \;;\; W^V = \left( \begin{array}{c|c} 0_{d_{\mathrm{x}} \times d_{\mathrm{x}}} & 0_{d_{\mathrm{x}} \times d_{\mathrm{y}}} \\ \hline W_0 & -I_{d_{\mathrm{y}}} \end{array} \right) \tag{26}$$

where $I_{d_{\mathrm{x}}} \in \mathbb{R}^{d_{\mathrm{x}} \times d_{\mathrm{x}}}$ and $I_{d_{\mathrm{y}}} \in \mathbb{R}^{d_{\mathrm{y}} \times d_{\mathrm{y}}}$ are identity matrices, $0_*$ are block matrices filled with zeros with the corresponding dimensions $*$, and $W_0 \in \mathbb{R}^{d_{\mathrm{y}} \times d_{\mathrm{x}}}$. In terms of the notation of Sec. 3.1, we have $d_{\mathrm{in}} = d_{\mathrm{out}} = d_{\mathrm{key}} = d_{\mathrm{x}} + d_{\mathrm{y}}$. For an input $x_t = [z_t, f(z_t)] \in \mathbb{R}^{d_{\mathrm{x}}+d_{\mathrm{y}}}$, this yields:

$$q_t = k_t = \left( \begin{array}{c} z_t \\ 0_{d_{\mathrm{y}} \times 1} \end{array} \right) \;;\; v_t = \left( \begin{array}{c} 0_{d_{\mathrm{x}} \times 1} \\ W_0 z_t - f(z_t) \end{array} \right) \tag{27}$$

With these key/value vectors, we can easily derive that, at time step $t$, the corresponding fast weight state is:

$$W_t = W_{t-1} + v_t \otimes k_t = W_{t-1} + \left( \begin{array}{c|c} 0_{d_{\mathrm{x}} \times d_{\mathrm{x}}} & 0_{d_{\mathrm{x}} \times d_{\mathrm{y}}} \\ \hline (W_0 z_t - f(z_t)) \otimes z_t & 0_{d_{\mathrm{y}} \times d_{\mathrm{y}}} \end{array} \right) \tag{28}$$

$$= \left( \begin{array}{c|c} 0_{d_{\mathrm{x}} \times d_{\mathrm{x}}} & 0_{d_{\mathrm{x}} \times d_{\mathrm{y}}} \\ \hline \sum_{\tau=1}^{t} (W_0 z_\tau - f(z_\tau)) \otimes z_\tau & 0_{d_{\mathrm{y}} \times d_{\mathrm{y}}} \end{array} \right) = \left( \begin{array}{c|c} 0_{d_{\mathrm{x}} \times d_{\mathrm{x}}} & 0_{d_{\mathrm{x}} \times d_{\mathrm{y}}} \\ \hline -\Delta W_t & 0_{d_{\mathrm{y}} \times d_{\mathrm{y}}} \end{array} \right) \tag{29}$$

---

where we can already recognize that $\Delta \boldsymbol{W}_t$ (in the left bottom block) is the gradients of the squared loss using $t$ inputs. The FWP output is:

$$\boldsymbol{y}_t = \boldsymbol{W}_t \boldsymbol{q}_t = \left( \begin{array}{c} \mathbf{0}_{d_x \times 1} \\ -\Delta \boldsymbol{W}_t \boldsymbol{z}_t \end{array} \right) \tag{30}$$

The last $d_y$ elements of output $\boldsymbol{y}_t$, which is $-\Delta \boldsymbol{W}_t \boldsymbol{z}_t$, is the same (up to a negative sign) to the linear model predictor trained by gradient descent as described above; a simple linear readout layer (with weight vector $\boldsymbol{w}^{\text{out}} = [\mathbf{0}_{d_x \times 1}, -\mathbf{1}_{d_y \times 1}]$) can easily apply the negative sign and extract this part of $\boldsymbol{y}_t$. Overall, at every step $t$, this FWP implements gradient descent using the corresponding data points up to $t$.

After having processed $T$ demonstration inputs, to make a prediction on a new $\boldsymbol{z}^\star$ without its target $f(\boldsymbol{z}^\star)$; we can set the corresponding "target" part of the model input as $\boldsymbol{x}^\star = [\boldsymbol{z}^\star, \boldsymbol{W}_0 \boldsymbol{z}^\star]$ to obtain $\boldsymbol{w}^{\text{out}\top} \boldsymbol{y}^\star = \Delta \boldsymbol{W}_T \boldsymbol{z}^\star$, corresponding to the gradient descent-trained linear predictor.

**Biology-compatible learning.**    The core idea of parameterizing local learning as part of the model's sequential dynamics (see Figure 2)—and metalearning the corresponding in-context learning algorithm—prompts us to rethink general research on biologically plausible learning in ANNs (Schmidhuber, 1989; 1990b; Mazzoni et al., 1991; O'Reilly, 1996; Bengio et al., 2015; Lillicrap et al., 2016; Pozzi et al., 2018; Najarro and Risi, 2020; Boopathy and Fiete, 2022; Hinton, 2022), which has been striving to address the longstanding critique of backpropagation in deep ANNs as being incompatible with biology (Crick, 1989; Zipser and Rumelhart, 1990) (e.g., a common critique is that backpropagation uses the transpose of the same weight matrix used in the forward pass in the backward pass, raising the "weight transport problem"). While meta-training still uses the biologically implausible algorithm (e.g., BPTT), once the model is trained, it becomes capable of in-context learning which is a local learning algorithm for which there is no obvious incompatibility with biology at a high level.

In particular, with FWPs in mind, one potentially fruitful view is, instead of drawing a parallel between the BPTT-based learning of slow weights with learning in the brain, we could introduce another time scale, and draw a parallel between local learning of fast weights with learning in the brain, and learning of slow weights could be analogous to evolution that has shaped our molecular mechanism and ability to learn (in fact, in lieu of BPTT, evolution strategy algorithms have also been used to train the slow weights of FWPs (Gomez and Schmidhuber, 2005); see also Chalmers (1990)).

As a side note, learning in ANNs has also received numerous critiques from cognitive scientists, who pointed out shortcomings such as the inability to learn from a few examples, to learn compositionally (Fodor and Pylyshyn, 1988), or to learn continually (McCloskey and Cohen, 1989). A recent series of work (Santoro et al., 2016; Lake and Baroni, 2023; Irie et al., 2025a) has addressed these classic challenges altogether through a common framework of metalearning and in-context learning (Irie and Lake, 2025).

### 3.6   Expressivity of the FWP models

In addition to the computational complexity, expressivity is a critical property to compare and categorize various types of sequence models in ML. Expressivity is about determining what types of computation the model can perform, which is a fundamental question in computer science, and is arguably also important in modeling cognitive abilities in psychology. One common misconception is that, given all the universal approximation and Turing completeness results available for these ANNs/RNNs (Siegelmann and Sontag, 1991; Pérez et al., 2019; Pérez et al., 2021; Orvieto et al., 2024), they are all equally powerful. This is not the case for practical models with finite resources; different sequence models differ in their "practical computational ability" (Weiss et al., 2018) and in the type of tasks they can solve.

**Tools to evaluate expressivity.**    Expressivity is often evaluated through formal language recognition tasks (Giles et al., 1989; Pollack, 1988; Gers and Schmidhuber, 2001; Schmidhuber et al., 2001; Weiss et al., 2018; Hahn, 2020; Merrill et al., 2020; Bhattamishra et al., 2020; Delétang et al., 2023; Irie et al., 2023; Merrill and Sabharwal, 2023; Strobl et al., 2024; Merrill et al., 2024; Beck et al., 2024). Formal languages are convenient

tools here because they provide us with a diverse set of tasks that require different types of sequence and memory processing abilities derived from the Chomsky hierarchy (Chomsky, 1956; Hopcroft and Ullman, 1969). For example, parity (given a sequence of 0s and 1s, the task of determining whether the number of 1s is odd) or modular arithmetic (addition and multiplication of integers modulo some integer) tasks can be represented as "regular languages" which require state-tracking ability to solve the task (Grazzi et al., 2025; Merrill et al., 2024; Sarrof et al., 2024), while certain "context-free" or "context-sensitive" grammars, such as the tasks of recognizing that a given sequence follows a pattern of $a^n b^n$ or $a^n b^n c^n$, can evaluate models' ability to count (Bhattamishra et al., 2020).

However, one important remark here is that the Chomsky hierarchy, which classifies theoretical models of computation, does not strictly capture the expressivity hierarchy of practical neural networks—the inability to solve certain regular language tasks does not imply a systematic failure on context-free grammar tasks, even though they are "stronger" than the regular languages in the Chomsky hierarchy. For example, this is exactly the case for the transformer: it either completely fails or struggles with certain regular languages, but their performance is excellent for both context-free and context-sensitive counting tasks (Bhattamishra et al., 2020).

**Expressivity of FWPs.** The FWP models in Table 1 differ in their expressive power. For this comparison, let's assume $d_{\text{out}} = d_{\text{in}} = d$, and regroup terms in the update rule equations that involve the current state $\boldsymbol{W}_{t-1}$, to obtain a canonical SSM-like form:

$$\boldsymbol{W}_t = \boldsymbol{B}_t \boldsymbol{W}_{t-1} \boldsymbol{A}_t + \boldsymbol{C}_t \tag{31}$$

for arbitrary matrices $\boldsymbol{A}_t, \boldsymbol{B}_t, \boldsymbol{C}_t \in \mathbb{R}^{d \times d}$ that have no dependency on variables from the previous time step $t-1$, where $\boldsymbol{A}_t$ and $\boldsymbol{B}_t$ are the "state transition matrices". For example, by denoting the identity matrix as $\boldsymbol{I} \in \mathbb{R}^{d \times d}$, the update rule of DeltaNet (see Eq. 24) can be rewritten as:

$$\boldsymbol{W}_t = \boldsymbol{W}_{t-1} + \psi(\beta_t)(\boldsymbol{v}_t - \boldsymbol{W}_{t-1}\phi(\boldsymbol{k}_t)) \otimes \phi(\boldsymbol{k}_t) \tag{24}$$

$$= \boldsymbol{W}_{t-1}\big(\boldsymbol{I} - \psi(\beta_t)\phi(\boldsymbol{k}_t) \otimes \phi(\boldsymbol{k}_t)\big) + \psi(\beta_t)\boldsymbol{v}_t \otimes \phi(\boldsymbol{k}_t) \tag{32}$$

that is, $\boldsymbol{A}_t = \big(\boldsymbol{I} - \psi(\beta_t)\phi(\boldsymbol{k}_t) \otimes \phi(\boldsymbol{k}_t)\big)$ for DeltaNet, which, as pointed out by Yang et al. (2024b), is a generalized Householder matrix (in Box 3, we provides an overview of the useful facts about the delta rule discussed in this work); and $\boldsymbol{B}_t = \boldsymbol{I}$.

More generally, the canonical form of Eq. 31 can tell a lot about the expressive power of the model by looking at the form that $\boldsymbol{A}_t$ and $\boldsymbol{B}_t$ take, because it is the state transition matrices which dictate the type of state transition the model can perform. The expressivity of models is limited when both $\boldsymbol{A}_t$ and $\boldsymbol{B}_t$ are reduced to an identity matrix (as is the case for the vanilla FWP; $\boldsymbol{A}_t = \boldsymbol{B}_t = \boldsymbol{I}$) or a diagonal matrix—whether all the diagonal values are the same (e.g., in RetNet, Mamba2, and mLSTM, $\boldsymbol{A}_t = \lambda \boldsymbol{I}$ or $\boldsymbol{A}_t = \lambda_t \boldsymbol{I}$) or different as in GLA ($\boldsymbol{A}_t = \boldsymbol{a}_t \otimes \boldsymbol{1} = \text{Diag}(\boldsymbol{a}_t)$), akin to element-wise recurrence (Sec. 2.2); see Table 1. For example, these diagonal state-transition-based models fail at recognizing certain regular languages such as parity and modular arithmetic, while DeltaNet models can handle such state-tracking tasks (Grazzi et al., 2025).

While earlier work on expressivity analysis of FWPs has been mostly empirical (Irie et al., 2021; 2023), there has been increasingly more work that aims to analyze and improve FWPs through the theoretical angle (see, e.g., Merrill et al. (2024); Sarrof et al. (2024); Muca Cirone et al. (2024); Movahedi et al. (2025)), in particular, Siems et al. (2025) introduced "DeltaProduct" which extends DeltaNet by using more than one application of the delta rule per time step, yielding $\boldsymbol{A}_t$ with a product of Householder matrices, achieving an improved expressivity in the resulting model.

In machine learning, the current challenge is to improve the expressivity and the general model performance, while maintaining the efficiency of sequence models (Yau et al., 2025). As a counterexample, introducing extra recurrence (by feeding back the output of fast net to the input of the slow net in the next time step (Irie et al., 2021)) or self-reference (by merging the slow and fast nets into a single network that modifies itself (Schmidhuber, 1992a; 1993a; Irie et al., 2022c)) can further improve the expressivity of FWPs, but it makes the models' training inefficient. However, more exploratory model developments by ignoring the requirement for efficient parallel training—which is desired solely from the machine learning standpoint, in principle—may also be fruitful for computational modeling in neuroscience.

Table 2: Complementarity of memory systems in machine learning. Reproduced from Irie et al. (2025b)

| Property | Transformer | Fast weight programmer |
|---|---|---|
| Complexity | quadratic | **linear** |
| Context length | bounded | **unbounded** |
| Retrieval precision | **high** | low |
| Expressivity | low | **high** (with certain update rules) |

### 3.7 Complementarity of memory systems

While recent developments of FWPs have produced efficient sequence models that are both more efficient and more expressive than the standard transformer—and competitive in practice on average across many language tasks (Yang et al., 2025; Siems et al., 2025; von Oswald et al., 2025)—the transformer still outperforms FWPs on precise retrieval tasks by a large margin (Irie et al., 2025b), suggesting the raison d'être of softmax attention. Their overall complementarity is summarized in Table 2. It remains an open question whether FWP models can be further improved to match the retrieval quality of standard transformers. For now, an engineering solution to achieve the best of both worlds is to combine the two in a hybrid architecture (Beck et al., 2024; Yang et al., 2025), which is reminiscent of the classic "complementary learning systems" (McClelland et al., 1995; O'Reilly and Norman, 2002) in which two complementary systems are allocated to collectively achieve incompatible goals that are unattainable by each individual system, through division of labor.

> **Box 3: Key facts and intuitions about the delta rule**
>
> Here we briefly summarize three useful facts about the delta rule.
>
> **1. Gradient descent on the squared error regression loss.** The delta rule equation can be derived from the following regression problem. Consider a function $\mathbb{R}^{d_{\text{in}}} \to \mathbb{R}^{d_{\text{out}}}$ parameterized by a matrix $\boldsymbol{W} \in \mathbb{R}^{d_{\text{out}} \times d_{\text{in}}}$, that transforms an arbitrary input $\boldsymbol{x} \in \mathbb{R}^{d_{\text{in}}}$ to output $f(\boldsymbol{W}\boldsymbol{x}) \in \mathbb{R}^{d_{\text{out}}}$ where $f$ is an arbitrary differentiable function $\mathbb{R}^{d_{\text{out}}} \to \mathbb{R}^{d_{\text{out}}}$. Given a data point with input $\boldsymbol{x} \in \mathbb{R}^{d_{\text{in}}}$ and target $\hat{\boldsymbol{y}} \in \mathbb{R}^{d_{\text{out}}}$ to which we want to fit this function, we can minimize the squared error $E(\boldsymbol{W}) = \frac{1}{2}||\hat{\boldsymbol{y}} - f(\boldsymbol{W}\boldsymbol{x})||_2^2$ between the model output and the target, using gradient descent.
>
> The corresponding gradient is: $\frac{\partial E}{\partial \boldsymbol{W}} = -\big(f'(\boldsymbol{W}\boldsymbol{x}) \odot (\hat{\boldsymbol{y}} - f(\boldsymbol{W}\boldsymbol{x}))\big) \otimes \boldsymbol{x}$, which yields the following update term to be added to $\boldsymbol{W}$ when one step of gradient descent is applied with a learning rate $\eta$: $\Delta \boldsymbol{W} = -\eta \frac{\partial E}{\partial \boldsymbol{W}} = \eta \big(f'(\boldsymbol{W}\boldsymbol{x}) \odot (\hat{\boldsymbol{y}} - f(\boldsymbol{W}\boldsymbol{x}))\big) \otimes \boldsymbol{x}$. In the case where the model is a simple linear layer, i.e., when $f$ is identity, this term becomes: $\Delta \boldsymbol{W} = \eta(\hat{\boldsymbol{y}} - \boldsymbol{W}\boldsymbol{x}) \otimes \boldsymbol{x}$, which corresponds to the delta rule used in DeltaNet, where the key $\phi(\boldsymbol{k}_t)$ and value $\boldsymbol{v}_t$ which play the role of input and target, respectively, and the learning rate $\eta_t$, are dynamically generated.
>
> **2. Improved update rule for a "key-value associative memory" system.** The delta rule can be seen as an improved update rule for a "key-value associative memory" system. Recall that a linear layer with an outer-product weight update rule can implement such a memory system—corresponding to Kohonen's correlation matrix memories (Kohonen, 1972) (in fact, Kohonen used the "key-data" terminology instead of "key-value"). Generally, the defining components of a memory architecture are its storage and the associated reading/writing primitives. In the case of a linear layer system, its weight matrix serves as the storage. The reading operation is the multiplication between a query input $\boldsymbol{q}$, and the memory matrix $\boldsymbol{W}$: $\boldsymbol{y} = \boldsymbol{W}\boldsymbol{q}$. A basic writing operation based on outer-product which adds a key-value association $(\boldsymbol{k}, \boldsymbol{v})$ to the memory is: $\boldsymbol{W}_t = \boldsymbol{W}_{t-1} + \boldsymbol{v} \otimes \boldsymbol{k}$. As an illustration, we consider 3-dimensional keys and 2-dimensional values. Given an empty memory $\boldsymbol{W}_0 = \boldsymbol{0}_{2 \times 3} \in \mathbb{R}^{2 \times 3}$, a key-value association, with an arbitrary value vector $\boldsymbol{v} \in \mathbb{R}^2$, and the key $\boldsymbol{k} = [0, 1, 0]^\top \in \mathbb{R}^3$ (which is a one-hot vector), can be stored to the memory by adding the corresponding outer-product to $\boldsymbol{W}_0$: $\boldsymbol{W}_1 = \boldsymbol{W}_0 + \boldsymbol{v} \otimes \boldsymbol{k} = [\boldsymbol{0}_{2 \times 1}; \boldsymbol{v}; \boldsymbol{0}_{2 \times 1}]$. We can retrieve the corresponding value by using the

corresponding key as the query $\boldsymbol{q} = [0, 1, 0]^\top$ for memory reading: $\boldsymbol{W}_1 \boldsymbol{q} = \boldsymbol{v}$. However, when the same association $(\boldsymbol{k}, \boldsymbol{v})$ is presented again to the system, the updated memory state becomes $\boldsymbol{W}_2 = [\boldsymbol{0}; 2\boldsymbol{v}; \boldsymbol{0}]$, which breaks the $(\boldsymbol{k}, \boldsymbol{v})$-association (the wrongly stored association is $(\boldsymbol{k}, 2\boldsymbol{v})$), as this naive additive rule does not check the current memory content. In contrast, the delta rule preserves the memory state in that case, as it only writes the "delta", i.e., the difference between the target value to be stored and the current value associated to the key.

**3. Improved transition matrix in linear RNNs.** Finally, as discussed in Sec. 3.6, the effect of the delta rule can also be understood in terms of a transition matrix in a linear RNN. The state transition matrix of the FWP with the purely additive rule is the identity matrix. In contrast, the delta rule introduces a state update term that is dependent on the current memory state; this yields a more expressive, non-diagonal transition matrix (Eq. 32).

**Glossary (Neuroscience)**

- **Ion channel:** A protein that forms a pore in the cell membrane, allowing specific ions (e.g., sodium $Na^+$ or calcium $Ca^{2+}$) to pass through. Ion channels are crucial for generating and transmitting electrical signals in neurons.

- **AMPA receptor:** ($\alpha$-amino-3-hydroxy-5-methyl-4-isoxazolepropionic acid receptor) A membrane protein (*ionotropic glutamate receptor*) that forms a *glutamate*-gated ion channel whose permeability depends on its subunit composition: receptors that include the GluA2 subunit are impermeable to $Ca^{2+}$, while those lacking GluA2 allow $Ca^{2+}$ entry. It opens rapidly and mediates fast excitatory transmission, contributing to synaptic plasticity.

- **NMDA receptor:** (N-methyl-D-aspartate receptor) A membrane protein (*ionotropic glutamate receptor*) that forms a glutamate- *and* voltage-gated ion channel permeable to $Na^+$, $K^+$, and $Ca^{2+}$. It requires depolarization to relieve a $Mg^{2+}$ block before opening; while $Na^+$ and potassium ($K^+$) also pass, the $Ca^{2+}$ influx is the principal signal that initiates synaptic plasticity.

- **Glutamate:** A small *amino acid neurotransmitter* that binds to receptors such as AMPA and NMDA, activating ion channels that mediate excitatory signaling and synaptic plasticity.

- **Phosphorylation:** An addition of a phosphate group to a protein or other molecule, typically by an enzyme called a kinase. Phosphorylation can alter a protein's activity, interactions, or localization.

- **Post-translational modification:** A chemical change to a protein after it is synthesized, such as phosphorylation. These modifications regulate protein function, stability, and signaling.

## 4 Neurobiology

Here we discuss how FWPs might be implemented in the brain (Sec. 4.1). This is necessarily speculative, though we will support our assertions with available evidence. In Sec. 4.2, we more broadly highlight properties of FWPs that are relevant as a synaptic plasticity model for neuroscience. Our hope is that these ideas will inspire new directions in the study of synaptic plasticity.

### 4.1 A neurobiological implementation of fast weight programming

To simplify the exposition, we will drop the time index and focus on the special case where the keys and queries are the same, and the values have a direct relation to the queries (as we specify below); thus, we have $\boldsymbol{q} = \boldsymbol{k}$. We consider a "postsynaptic" population of neurons that receive "presynaptic" input **q** and generate firing rates **y**. The presynaptic neurons receive input from a sensory representation **x**, which we leave implicit here. As in many models of neural activity, we will assume that the postsynaptic firing rate can be approximated by a linear combination of presynaptic inputs, $y_j = \sum_i W_{ji} q_i$.

The synaptic strengths evolve according to a generalized Hebbian learning rule, where strength increases due to the coincidence of presynaptic firing and a postsynaptic activity trace:

$$\Delta W_{ji} \propto v_j k_i, \tag{33}$$

where $k_i = q_i$ is the firing rate of presynaptic neuron $i$, and $v_j$ is an "activity trace" encoded by postsynaptic neuron $j$, which we assign to the accumulated postsynaptic calcium level. We model the calcium trace as a linear combination of the presynaptic inputs, $v_j = \sum_i U_{ji} q_i$ (though in reality the relationship is nonlinear due to the voltage-dependence of calcium conductance; e.g., we can possibly introduce some non-linear activation function on $\boldsymbol{v}$). This is equivalent to the FWP setup with $\boldsymbol{W}^V = \boldsymbol{U}\boldsymbol{W}^Q$ and $\boldsymbol{W}^Q = \boldsymbol{W}^K$.

We hypothesize that the synaptic strength matrix $\boldsymbol{W}$ corresponds to the density/conductance of *AMPA receptors*, while the matrix $\boldsymbol{U}$ governing the calcium response corresponds to the density/conductance of *NMDA receptors*. This distinction is motivated by several facts. First, firing rates are primarily governed by sodium *channels* linked to AMPA receptors, whereas intracellular calcium levels are primarily governed by calcium channels linked to NMDA receptors. Second, AMPA receptor plasticity can be induced quickly (on the order of seconds (Gustafsson et al., 1989)) by Hebbian stimulation protocols—fast enough to contribute to performance (at least in principle) on working memory tasks (Erickson et al., 2010; Lansner et al., 2023). In contrast, induction of NMDA receptor plasticity is typically slower (Hunt and Castillo, 2012). These two forms of plasticity can also be induced independently. Third, AMPA receptor plasticity critically depends on calcium influx, which activates a cascade of protein synthesis and *post-translational modification*. This is consistent with the dependence of fast weight modification on $v_j$, the putative calcium trace. In particular, the fastest change induced by Hebbian stimulation is the *phosphorylation* of AMPA receptors by calcium-activated kinases like PKA and CaMKII (Lee et al., 2000; Soderling and Derkach, 2000).

The generalizations of the vanilla FWP architecture (discussed in Sec. 3.4) suggest further nuances to this picture. For example, DeltaNet uses $v_j - \sum_i W_{ji} q_i$ in place of $v_j$ in the Hebbian update. A possible biological interpretation is that recent neural activity sets a plasticity threshold on postsynaptic calcium, reversing the direction of plasticity when calcium levels are below the threshold. Indeed, this is a venerable idea in models of synaptic plasticity (Shouval et al., 2002; Graupner and Brunel, 2012).

Another generalization discussed in Sec. 3.4 (see also Table 1) is the use of various forms of decay on the weights and/or states. For example, RetNet assumes scalar decay of all fast weights. This is broadly consistent with the observation that changes to synaptic strength continuously decay due to a variety of processes (e.g., molecular turnover, diffusion of synaptic components, stochastic kinase/phosphatase activity). Decay is also controlled by homeostatic processes that seek to keep neural activity near a set point (Turrigiano, 2008). Mamba2 assumes that decay is input-dependent, which is broadly consistent with the role of activity-dependent mechanisms in determining the decay rate of synaptic plasticity (Abraham and Williams, 2003). In particular, protein synthesis triggered by calcium influx plays a central role in the conversion of short-term synaptic changes (e.g., AMPA receptor phosphorylation) to long-term changes (trafficking of new AMPA receptors to the postsynaptic membrane). GLA further extends this input-dependent decay by modeling separate decay rates for each postsynaptic neuron.

The idea that synaptic plasticity plays out at multiple timescales through several different mechanisms has become widely accepted (Citri and Malenka, 2008), including some mechanisms (such as short-term depression and facilitation) that are even faster than the fast Hebbian plasticity described earlier (Zador and Dobrunz, 1997). Our goal in this section was to link a subset of these mechanisms to the computational ideas underlying FWP systems. This leaves fertile ground for future exploration of other potential links, including transformer-like computation in the brain (Ellwood, 2024; Whittington et al., 2022; 2025; Kozachkov et al., 2023; Gershman et al., 2025).

### 4.2 Other prospects and considerations for neuroscience

More broadly, the base FWP equation presented here (Eqs. 13-14) may be extended (or restricted) to accommodate certain neurobiological aspects that are not supported by the conventional computational models in neuroscience. In particular, the following properties are prominent.

First, FWPs could implement a quite broad class of synaptic modifications (Magee and Grienberger, 2020), including both Hebbian and non-Hebbian ones, by conceiving extensions in which the key, value, and query representations come from independent sources (i.e., they do not all have to be a function of the shared input $\boldsymbol{x}_t$) or even from different time steps. In particular, FWPs can naturally support non-Hebbian learning, where the synaptic weight modification does not depend on the postsynaptic firings. For example, behavioral

timescale synaptic plasticity (BTSP; Bittner et al. (2017); Wu and Maass (2025)) in the hippocampus of mammals is well known to be non-Hebbian (i.e., it does not depend on the input-output correlation). BTSP involves different sub-regions of the hippocampus, CA1 and CA3, and a part of entorhinal cortex called EC3. Their functional relationship can be parameterized as an FWP, in which the fast network maps input CA3 activities (query $\boldsymbol{q}_t$) to output CA1 activations (output $\boldsymbol{y}_t$), whose synaptic weights are modulated by CA3 (key $\boldsymbol{k}_t$) and gated by EC3 (value $\boldsymbol{v}_t$)—note that an outer product, like other products, can implement a gate. The FWP framework also supports the cases where either or both the slow or fast networks in such a system employ recurrent connectivity (Irie et al., 2021). In contrast, to implement Hebbian learning, FWPs may use variables from previous time steps as keys and values (as is done in certain recurrent FWPs (Schmidhuber, 1993b)). Overall, the FWP framework provides a unified formalism for modeling synaptic plasticity across types and timescales, which can facilitate the development of computational models in neuroscientific studies (cf., e.g., Aitken and Mihalas (2023)).

Second, unlike traditional auto-associative memory models (Anderson, 1970; Amari, 1972; Nakano, 1972) in neuroscience which focus on retrieval of clean patterns based on partially corrupted patterns (Amari, 1972; Hopfield, 1982) (see also Kanerva (1988); Millidge et al. (2022)), FWPs implement flexible hetero-associative memory, which is functionally more general (Kohonen, 1972; Steinbuch, 1961); as long as the query-key matching function is discriminative enough, it allows to store arbitrary key-value associations, regardless of whether the value is a denoised version of the key (for auto-association) or any other arbitrary patterns.

Finally, the current form of FWPs designed for machine learning purposes could be extended to include additional properties that are currently missing from a neurobiological perspective, such as stochasticity or the time window for plasticity—an important characteristic that distinguishes different types of synaptic plasticity. We hope this Primer provides a solid foundation that inspires future exploration of such potential extensions.

## 5 Conclusion

The main goal of this Primer was to introduce the concept of fast weight programmers (FWPs)—a special class of recurrent neural networks (RNNs) with two-dimensional hidden states—at the nexus of machine learning and computational neuroscience.

We have highlighted unique properties of FWPs that are relevant from various perspectives in these fields. We have argued that the use of dynamically changing synaptic weights as a form of short-term memory offers a compelling abstract computational model for synaptic plasticity, capturing timescales that traditional RNNs with static weights cannot. In machine learning, such sequential dynamics have been playing a central role in developing modern sequence models, as they allow for both sequence-level parallelism—crucial for efficient training, and therefore, scalability—and more expressive computations than those supported by the now popular transformer. Furthermore, the ability of FWPs to intuitively express local learning—that is, learning that only involves locally available variables—within their own sequential dynamics through weight/state update rules provides a novel perspective and a promising framework for learning mechanisms compatible with biological constraints.

Finally, we have also explored a neurobiological implementation of FWP-like computations in the brain, and broadly discussed the FWP concept as a general and promising framework that supports various types of synaptic plasticity rules that are known in neuroscience, including both Hebbian and non-Hebbian rules. While highly speculative and preliminary, we hope this work opens new avenues for modeling synaptic modulation and for discussing their roles in learning and memory in the brain.

## Acknowledgments

The authors are grateful for support from the Kempner Institute for the Study of Natural and Artificial Intelligence, a Polymath Award from Schmidt Sciences, and the Department of Defense MURI program under ARO grant W911NF-23-1-0277. Kazuki Irie thanks Imanol Schlag and Jürgen Schmidhuber for introducing him to the world of fast weights, while at the Swiss AI lab IDSIA.

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

## A Derivations connecting the update rules and local losses

Here we provide derivations connecting the update rules and the loss functions provided in Table 1. In the following, **we omit the activation function $\phi$ on keys** (i.e., we replace $\phi(\boldsymbol{k}_t)$ by $\boldsymbol{k}_t$), which does not play any role in the derivations w.r.t. $\boldsymbol{W}$. Here we also do not specify any dimensions and instead work with arbitrary ones including ranges for the running indices in the sums. $i, j, l, m$ denote positive integers.

**Vanilla FWP.** The local loss is defined as a similarity term:

$$\mathcal{L}_t(\boldsymbol{W}) = -\boldsymbol{v}_t^\top \boldsymbol{W} \boldsymbol{k}_t = -\sum_l \boldsymbol{v}_{t|l} \sum_m \boldsymbol{W}_{l,m} \boldsymbol{k}_{t|m} \tag{34}$$

where $\boldsymbol{k}_{t|m} \in \mathbb{R}$ denotes the $m$-th element of vector $\boldsymbol{k}_t$; we use the notation $|$ to clearly separate the time index $t$ from the coordinate index $m$.

By taking the derivative w.r.t. an element $\boldsymbol{W}_{i,j} \in \mathbb{R}$ of matrix $\boldsymbol{W}$, we obtain:

$$\frac{\partial \mathcal{L}_t}{\partial \boldsymbol{W}_{i,j}}(\boldsymbol{W}) = -\boldsymbol{v}_{t|i} \boldsymbol{k}_{t|j} \tag{35}$$

which yields the matrix form with an outer product: $\frac{\partial \mathcal{L}_t}{\partial \boldsymbol{W}}(\boldsymbol{W}) = -\boldsymbol{v}_t \otimes \boldsymbol{k}_t$. Using a learning rate of 1, one step of gradient descent corresponds to the weight update:

$$\boldsymbol{W}_t = \boldsymbol{W}_{t-1} - \frac{\partial \mathcal{L}_t}{\partial \boldsymbol{W}}(\boldsymbol{W}_{t-1}) = \boldsymbol{W}_{t-1} + \boldsymbol{v}_t \otimes \boldsymbol{k}_t \tag{36}$$

**DeltaNet.** The local loss is the least square between a target $\boldsymbol{v}_t$ and the current "net output" $\boldsymbol{W} \boldsymbol{k}_t$:

$$\mathcal{L}_t(\boldsymbol{W}) = \frac{1}{2}\|\boldsymbol{v}_t - \boldsymbol{W} \boldsymbol{k}_t\|_2^2 = \frac{1}{2}\sum_l \left(\boldsymbol{v}_{t|l} - \sum_m \boldsymbol{W}_{l,m} \boldsymbol{k}_{t|m}\right)^2 \tag{37}$$

The derivative is:

$$\frac{\partial \mathcal{L}_t}{\partial \boldsymbol{W}_{i,j}}(\boldsymbol{W}) = -(\boldsymbol{v}_{t|i} - \sum_m \boldsymbol{W}_{i,m} \boldsymbol{k}_{t|m})\boldsymbol{k}_{t|j} \tag{38}$$

which yields the matrix form:

$$\frac{\partial \mathcal{L}_t}{\partial \boldsymbol{W}}(\boldsymbol{W}) = -(\boldsymbol{v}_t - \boldsymbol{W} \boldsymbol{k}_t) \otimes \boldsymbol{k}_t \tag{39}$$

Using a learning rate of $\eta_t$, one step of gradient descent yields:

$$\boldsymbol{W}_t = \boldsymbol{W}_{t-1} - \eta_t \frac{\partial \mathcal{L}_t}{\partial \boldsymbol{W}}(\boldsymbol{W}_{t-1}) = \boldsymbol{W}_{t-1} + \eta_t(\boldsymbol{v}_t - \boldsymbol{W}_{t-1}\boldsymbol{k}_t) \otimes \boldsymbol{k}_t \tag{40}$$

Note that this derivation essentially corresponds to how Widrow and Hoff (1960) derived the delta rule.

**OjaNet.** The local objective is defined as the sum of the similarity loss as in the vanilla FWP case with an additional constraint term:

$$\mathcal{L}_t(\boldsymbol{W}) = -\boldsymbol{v}_t^\top \boldsymbol{W} \boldsymbol{k}_t + \frac{1}{2}\|\boldsymbol{W}^\top \boldsymbol{v}_t\|_2^2 = -\sum_l \boldsymbol{v}_{t|l} \sum_m \boldsymbol{W}_{l,m} \boldsymbol{k}_{t|m} + \frac{1}{2}\sum_m \left(\sum_l \boldsymbol{W}_{l,m} \boldsymbol{v}_{t|l}\right)^2 \tag{41}$$

The derivative is:

$$\frac{\partial \mathcal{L}_t}{\partial \boldsymbol{W}_{i,j}}(\boldsymbol{W}) = -\boldsymbol{v}_{t|i} \boldsymbol{k}_{t|j} + \boldsymbol{v}_{t|i}\left(\sum_l \boldsymbol{W}_{l,j} \boldsymbol{v}_{t|l}\right) \tag{42}$$

which yields the matrix form:

$$\frac{\partial \mathcal{L}_t}{\partial \boldsymbol{W}}(\boldsymbol{W}) = -\boldsymbol{v}_t \otimes \left(\boldsymbol{k}_t - \boldsymbol{W}^\top \boldsymbol{v}_t\right) \tag{43}$$

Using a learning rate of $\eta_t$, one step of gradient descent yields:

$$\boldsymbol{W}_t = \boldsymbol{W}_{t-1} - \eta_t \frac{\partial \mathcal{L}_t}{\partial \boldsymbol{W}}(\boldsymbol{W}_{t-1}) = \boldsymbol{W}_{t-1} + \eta_t \boldsymbol{v}_t \otimes (\boldsymbol{k}_t - \boldsymbol{W}_{t-1}^\top \boldsymbol{v}_t) \tag{44}$$

This corresponds to Oja's rule (Oja, 1982) by treating $\boldsymbol{v}_t$ as the net output $\boldsymbol{W}_{t-1}\boldsymbol{k}_t$ (the same way we treat the first term as a Hebbian-like term). The corresponding loss above directly parallels Oja's objective of stabilizing the Hebbian rule by preserving the norm of the weight vector—i.e., enforcing a unit-norm constraint in the single-neuron case, which in our matrix formulation generalizes to a row-wise orthonormality constraint: $\boldsymbol{W}\boldsymbol{W}^\top = \boldsymbol{I}$. The additional quadratic term in our loss, $\frac{1}{2}||\boldsymbol{W}^\top \boldsymbol{v}_t||_2^2$, serves the same purpose as it introduces the same correction term that keeps $\boldsymbol{W}\boldsymbol{W}^\top \approx \boldsymbol{I}$. This equivalence can be shown formally by solving the constrained optimization problem subject to $\boldsymbol{W}\boldsymbol{W}^\top = \boldsymbol{I}$, introducing Lagrange multipliers, and recovering the same update rule.

**State Decay Variants.** The local loss function for all the state decaying variants correspond to the similarity loss as in the vanilla FWP case with an additional $L_2$ regularization term on the fast weight matrix. Different variants differ from each other in how the $L_2$ term is weighted/scaled.

For example for **RetNet**, the local loss function is:

$$\mathcal{L}_t(\boldsymbol{W}) = -\boldsymbol{v}_t^\top \boldsymbol{W} \phi(\boldsymbol{k}_t) + \frac{1-\lambda}{2}||\boldsymbol{W}||_F^2 = -\sum_l \boldsymbol{v}_{t|l} \sum_m \boldsymbol{W}_{l,m} \boldsymbol{k}_{t|m} + \frac{1-\lambda}{2} \sum_l \sum_m \boldsymbol{W}_{l,m}^2 \tag{45}$$

The derivative is:

$$\frac{\partial \mathcal{L}_t}{\partial \boldsymbol{W}_{i,j}}(\boldsymbol{W}) = -\boldsymbol{v}_{t|i} \boldsymbol{k}_{t|j} + (1-\lambda)\boldsymbol{W}_{i,j} \tag{46}$$

which yields the matrix form:

$$\frac{\partial \mathcal{L}_t}{\partial \boldsymbol{W}}(\boldsymbol{W}) = -\boldsymbol{v}_t \otimes \boldsymbol{k}_t + (1-\lambda)\boldsymbol{W} \tag{47}$$

Using a learning rate of 1, one step of gradient descent yields:

$$\boldsymbol{W}_t = \boldsymbol{W}_{t-1} - \frac{\partial \mathcal{L}_t}{\partial \boldsymbol{W}}(\boldsymbol{W}_{t-1}) = \boldsymbol{W}_{t-1} + \boldsymbol{v}_t \otimes \boldsymbol{k}_t - (1-\lambda)\boldsymbol{W}_{t-1} = \lambda\boldsymbol{W}_{t-1} + \boldsymbol{v}_t \otimes \boldsymbol{k}_t \tag{48}$$

The derivation is analogous for **Mamba2**, **xLSTM**, and **Gated RFA**.

The case of **GLA** is worth its own derivation as its formula looks slightly more complex as different scales (elements of $\boldsymbol{a}_t$) are used for different rows of matrix $\boldsymbol{W}$. Its loss function is:

$$\frac{\partial \mathcal{L}_t}{\partial \boldsymbol{W}}(\boldsymbol{W}) = -\boldsymbol{v}_t^\top \boldsymbol{W} \phi(\boldsymbol{k}_t) + \frac{1}{2}||((\sqrt{1-\boldsymbol{a}_t}) \otimes \mathbf{1}) \odot \boldsymbol{W}||_F^2 \tag{49}$$

$$= -\sum_l \boldsymbol{v}_{t|l} \sum_m \boldsymbol{W}_{l,m} \boldsymbol{k}_{t|m} + \frac{1}{2} \sum_l \sum_m \left((1-\boldsymbol{a}_{t|l})\boldsymbol{W}_{l,m}\right)^2 \tag{50}$$

where $(1-\boldsymbol{a}_t)$ denotes a vector of the same size as $\boldsymbol{a}_t$ whose entries are $1 - \boldsymbol{a}_{t|i}$ for all $i$.

The derivative is:

$$\frac{\partial \mathcal{L}_t}{\partial \boldsymbol{W}_{i,j}}(\boldsymbol{W}) = -\boldsymbol{v}_{t|i} \boldsymbol{k}_{t|j} + (1-\boldsymbol{a}_{t|i})\boldsymbol{W}_{i,j} \tag{51}$$

which yields the matrix form:

$$\frac{\partial \mathcal{L}_t}{\partial \boldsymbol{W}}(\boldsymbol{W}) = -\boldsymbol{v}_t \otimes \boldsymbol{k}_t + \big((1 - \boldsymbol{a}_t) \otimes \boldsymbol{1}\big) \odot \boldsymbol{W} \tag{52}$$

Using a learning rate of 1, one step of gradient descent yields:

$$\boldsymbol{W}_t = \boldsymbol{W}_{t-1} - \frac{\partial \mathcal{L}_t}{\partial \boldsymbol{W}}(\boldsymbol{W}_{t-1}) = \boldsymbol{W}_{t-1} + \boldsymbol{v}_t \otimes \boldsymbol{k}_t - \big((1 - \boldsymbol{a}_t) \otimes \boldsymbol{1}\big) \odot \boldsymbol{W}_{t-1} \tag{53}$$

$$= (\boldsymbol{a}_t \otimes \boldsymbol{1}) \odot \boldsymbol{W}_{t-1} + \boldsymbol{v}_t \otimes \boldsymbol{k}_t \tag{54}$$

**Gated DeltaNet.** The Gated DeltaNet case can be straightforwardly obtained by combining the derivations of the DeltaNet case and the state decay case above, and using a learning rate $\eta_t$.

