# OpenReview forum: "Fast weight programming and linear transformers: from machine learning to neurobiology"
_TMLR — Accepted by TMLR_

### Review · Reviewer_9Co4 · 2025-10-18

**Summary Of Contributions:**

This paper gives a primer on Fast Weight Programmers (FWPs), an advancement over RNNs, which maintains a memory matrix instead of a vector in RNNs. The paper goes on to draw connections between linear transformers and FWPs highlighting that FWPs have high expressivity while bein efficient, yet are harder to train. Further the authors also draw connections with neurobiology.

The paper is a very well written review on the existing methods in this space Linear RNNs and SSMs, yet does not introduce any new insight, but rather focuses on drawing connections between existing ideas.

Small correction, in the first figure, the key and query are interchanged. Attention is between key and query.

**Audience:**

Yes

**Audience Explanation:**

Readers who are not familiar with this line of work might benefit from this work.

**Broader Impact Concerns:**

None.

**Claims And Evidence:**

Yes

**Claims Explanation:**

The paper is a review of existing concepts, highlighting their potential connections.

**Requested Changes:**

Fix the typo in Fig 1.

---

> ### Author Response · Authors · 2025-10-18
> **Author Response**
>
> We thank the reviewer for their valuable comments and for the positive feedback on our writing.
>
> Indeed, this is a primer/review paper---all the mathematical concepts and connections are based on prior work (as we clarify upfront in the abstract and introduction). We believe this synthesis alone constitutes a useful contribution to TMLR and the broader machine learning community, especially given the increasing interest in these models for building large language models, see, e.g., Alibaba's Qwen3-Next released last month (September 2025).
>
> In addition, this primer aims to introduce these machine learning concepts of fast weight programming to the neuroscience community, by discussing their potential relevance as computational models for neurobiology (Sec 4)---this perspective represents a truly unique contribution of this work.
>
> Finally, regarding Figure 1, thank you for your comment. However, that is not a typo. "Attention" is a function whose arguments are keys, values, and a query (see Eq. 9); Figure 1 illustrates this dependency (arrows) by showing all these variables as inputs (without going into the details of the internal computations inside the Attention box). Since the goal of Figure 1 is to contrast the three models at this level of abstraction, we plan to keep the current figure as is---we nevertheless thank you for pointing this out.
>
> We hope this further clarifies our contributions and addresses your comment.

---

### Review · Reviewer_HYzD · 2025-10-24

**Summary Of Contributions:**

First of all, thank you for writing this work, I really enjoyed reading it.
The paper presents the Fast Weight Programming (FWP) framework and discusses how it relates to existing recurrent neural networks (RNNs) and transformers. In particular, the authors show that both state-space models and linear transformers can be expressed using the FWP equations, hence providing a unifying notation that helps bridge these architectures.
In the second part, the paper explores potential extensions of FWP to model computational processes in the brain, though these discussions remain highly speculative.

Strengths
• The paper is well written and the connection with transformers is clear and easy to follow.
• The idea of framing different existing formulations (RNNs, state-space models, transformers) under a unifying mathematical notation is excellent. It provides conceptual clarity, helps highlight commonalities, and facilitates comparison of models in terms of novelty and usefulness.

Weaknesses
1. Connection to RNNs less clear.
The link between FWP and standard RNNs is less straightforward than the transformer case. Conceptually, the idea of a 2D hidden state in FWP makes sense, but in a standard RNN, the hidden-to-hidden transformation W_h*h_t is fixed. In FWP, W_t appears to replace this product, but it would help to clarify this equivalence explicitly.
Also, what are the direct analogs of q_t, k_t, and v_t in the standard RNN formulation?
2. Terminology: "Fast" may be misleading.
The term fast only makes sense when the "slow" network learns via a slower algorithm such as backpropagation through time (BPTT). However, if the slow network were trained using an online learning rule, the distinction in timescale might disappear.
Moreover, during inference (when slow weights are fixed), referring to them as "slow-learning" weights is technically inaccurate. The terminology would make more sense in a continual learning context.
3. Naming convention.
The phrase "training a network to generate the weights of another network" aligns more closely with the established hypernetwork definition than FWP. "Fast Weight Programming" may not fully capture this relationship (though I recognise this is naming convention).
4. Speculative neuroscience connections.
The section "Other prospects and considerations for neuroscience" draws interesting but highly speculative parallels between FWP and the CA3–CA1 hippocampal circuitry. The analogy is not fully accurate, as CA3 is known to be recurrently connected while the slow weights in FWP are not.

**Audience:**

Yes

**Audience Explanation:**

The topic is highly relevant to both the machine learning and computational neuroscience communities, and it will likely attract considerable interest from readers working at the intersection of these areas.

**Broader Impact Concerns:**

No concerns.

**Claims And Evidence:**

Yes

**Claims Explanation:**

Most claims are clear and supported by accurate reasoning. However, some statements lack supporting evidence.
For example, in the Practical considerations paragraph:
"(e.g., we found the opposite trend for reinforcement learning in certain game environments); general weight/memory decay may hurt in certain scenarios."
There are no results or references provided to support this claim (though I understand this is a Primer rather than an empirical paper).
Additionally, the comment in the final paragraph of Section 2.1 might not hold under online or real-time recurrent learning scenarios, and this could be briefly acknowledged.

**Requested Changes:**

Mainly address the weaknesses listed above and provide clarification or evidence where appropriate.
Among them, point (2) regarding the definition and interpretation of "fast" vs "slow" learning is the most critical for strengthening the conceptual clarity and supporting an acceptance recommendation.
The remaining points would further improve the paper’s precision and readability.

---

> ### Author Response · Authors · 2025-10-25
> **Author Response Part 1/2**
>
> We thank the reviewer for the valuable and thorough comments, as well as for the very positive feedback about the relevance of this paper.
>
> We would like to address the remaining concerns (related ones are grouped together) as follows:
>
> **Weaknesses**
>
> > 1. Connection to RNNs less clear.
>
> First of all, unlike with transformers (Sec 3.3), we did not claim any mathematical equivalence between any specific RNN model and any specific FWP model; instead, we characterize them as sharing a common RNN functional form. The canonical form of an RNN is $s_t = f(s_t-1, x_t)$ where $s_t$ and $s_{t-1}$ are updated and old RNN states respectively, and $x_t$ is the current input; and f is the transition function. The vanilla FWP of Sec 3.1 also follows this form, with $W_t$ as the hidden state, and the transition function f encapsulates the mapping from $x_t$ to $k_t$ and $v_t$ as well as their usage to update the state.
>
> A slightly more formal connection can be drawn with the linear RNNs (a.k.a. state space models; SSM) as vanilla FWP’s Eq (13) is precisely a linear RNN equation with 2D hidden state and the identity matrix as the transition matrix (we may even flatten the 2D state to express it as a 1D vector-state RNN); and the query part corresponds to the output readout ($y_t = W_{out} s_t$ in SSMs vs. $y_t = W_t q_t$ in FWPs). Given that in the current version, there is only one sentence about this connection in Sec 2.2. (*“By recognizing that such a definition can also express models in which st is a matrix instead of a vector, we will see a connection to fast weight programmers (Sec. 3).”*), we will add an extra sentence to clarify this in Sec. 3.1 in the revision.
>
> > 2. Terminology: "Fast" may be misleading.
>
> > Additionally, the comment in the final paragraph of Section 2.1 might not hold under online or real-time recurrent learning scenarios, and this could be briefly acknowledged.
>
> > point (2) regarding the definition and interpretation of "fast" vs "slow" learning is the most critical for strengthening the conceptual clarity and supporting an acceptance recommendation.
>
> Thank you for pointing this out. The reviewer is right that when a *fully* online learning algorithm (i.e., an online sequence learning algorithm with update frequency of **one time step**, e.g., real-time recurrent learning (RTRL) with step 1) is used, the “fast” vs. “slow” timescale distinction of FWP may disappear. However, such a fully online learning of “slow weights” rather remains a theoretical edge case, given that, in practice RTRL with frequent updates often invalidates accumulated sensitivities based on outdated weights and typically leads to unstable or inefficient learning dynamics (see, e.g., a recent work uses at least 10 steps [1])---so learning of “slow weights” inherently involves a “slow-learning” algorithm.
>
> Nevertheless, following the reviewer’s suggestion, we’ll add a sentence acknowledging this special case. We thank the reviewer once again for pointing this out.
>
> Accordingly, while we acknowledge that the “slow-learning” descriptor is not literally applicable during inference, we believe the terminology (which we did not invent---following the historical work on fast weights dating back to 1980s/90s) remains conceptually and practically appropriate for describing the characteristic timescale difference underlying FWPs.
>
> > 3. Naming convention. The phrase "training a network to generate the weights of another network" aligns more closely with the established hypernetwork definition than FWP.
>
> It is true that the hypernetwork paper (ICLR 2017) [2] has successfully popularized the concept of networks generating weights for another network (as note in the second paragraph of Sec. 3.2). However, as acknowledged by the hypernetwork paper itself, the concept of fast weights predates it: *“Even before the work on HyperNEAT and DCT, Schmidhuber (1992; 1993) has suggested the concept of fast weights in which one network can produce context-dependent weight changes for a second network.”* (see their related work section).
>
> The weight generation concept was the essence of fast weight programmers: a follow-up work [3] proposed the concept of a “network generating weights *for itself*" as a natural extension of FWP. More recently, this weight generation view precisely motivated certain modern extensions of FWP such as DeltaNet, as well as conception of in-context learning [4].

---

> ### Author Response · Authors · 2025-10-25
> **Author Response Part 2/2**
>
> > 4. Speculative neuroscience connections. The section "Other prospects and considerations for neuroscience" draws interesting but highly speculative parallels between FWP and the CA3–CA1 hippocampal circuitry. The analogy is not fully accurate, as CA3 is known to be recurrently connected while the slow weights in FWP are not.
>
> Thank you for pointing this out. The strength of our proposal here precisely lies in the fact that we did not introduce a single fixed model but rather a *framework* that defines a broader model hypothesis space. The FWP model family also includes models with a recurrent slow network (which is precisely called “recurrent FWP” [5]). We will add the corresponding comment in the revision. If the reviewer believes there are additional relevant references we should include, we would be happy to do so.
>
> **Claim accuracy**
>
> > For example, in the Practical considerations paragraph: "(e.g., we found the opposite trend for reinforcement learning in certain game environments); general weight/memory decay may hurt in certain scenarios." There are no results or references provided to support this claim (though I understand this is a Primer rather than an empirical paper).
>
> Unfortunately, the corresponding results are currently unpublished practitioner knowledge. If this is problematic, we are happy to remove the corresponding sentence. Instead, we will add a more general recommendation to avoid relying solely on existing results that may be overfitted to the development of language models when applying FWPs as general-purpose sequence models to other settings. In fact, conceptually, it is reasonable to expect that as sequence lengths increase, mechanisms such as decay terms that systematically erase older memories over time can be problematic in certain tasks (e.g., image generation [6]).
>
> We hope our responses above have addressed all the reviewer’s remaining concerns. We will upload a revised PDF soon.
>
> ------------------------
>
> **References** (all these references are cited in the paper)
>
> [1] Kazuki Irie, Anand Gopalakrishnan, and Jürgen Schmidhuber. Exploring the promise and limits of real-time recurrent learning. In Int. Conf. on Learning Representations (ICLR), Vienna, Austria, May 2024.
>
> [2] David Ha, Andrew Dai, and Quoc V Le. Hypernetworks. In Int. Conf. on Learning Representations (ICLR), Toulon, France, April 2017.
>
> [3] Jürgen Schmidhuber. A self-referential weight matrix. In Proc. Int. Conf. on Artificial Neural Networks (ICANN), pages 446–451, Amsterdam, Netherlands, September 1993b.
>
> [4] Johannes von Oswald, Eyvind Niklasson, Ettore Randazzo, João Sacramento, Alexander Mordvintsev, Andrey Zhmoginov, and Max Vladymyrov. Transformers learn in-context by gradient descent. In Proc. Int. Conf. on Machine Learning (ICML), Honolulu, HI, USA, July 2023b.
>
> [5] Kazuki Irie, Imanol Schlag, Róbert Csordás, and Jürgen Schmidhuber. Going beyond linear transformers with recurrent fast weight programmers. In Proc. Advances in Neural Information Processing Systems (NeurIPS), Virtual only, December 2021.
>
> [6] Kazuki Irie and Jürgen Schmidhuber. Images as weight matrices: Sequential image generation through
> synaptic learning rules. In Int. Conf. on Learning Representations (ICLR), Kigali, Rwanda, May 2023

---

> > ### Comment · Reviewer_HYzD · 2025-11-05
> > **reply**
> >
> > Thank you, we are happy with the detailed reply by the authors and the alterations.

---

### Review · Reviewer_JBY1 · 2025-10-28

**Summary Of Contributions:**

This work provides a review of Fast Weight Programmers (FWPs) and establishes its formal connections with state space models, recurrent neural networks and transformers. The authors explore variations of FWPs found in existing literature and provide practical considerations for training these models. The paper emphasizes how FWPs capture learning dynamics through local online optimization, with update rules corresponding to specific local objective functions. Additionally, the authors examine the relationship between FWPs and meta-learning/in-context learning, and reflect on two formulations of meta-learning as sequence learning problems: the delayed-feedback setting and the synchronous-feedback setting. Furthermore, they propose viewing fast-weight learning in FWPs as analogous to learning in the brain, while slow-weight learning corresponds to evolution. They also review the use of formal languages for evaluating expressivity and discusses how expressivity differences among FWP variants are influenced by model architectures. Lastly, the paper proposes a neurobiological implementation of FWPs where the fast weights are influenced by the AMPA receptors, the slow weights are influenced by NMDA receptor and suggests that the FWP could implement both Hebbian and non-Hebbian synaptic modifications. Overall, the paper provides a comprehensive and accessible introduction to FWPs that bridges machine learning and neuroscience, making it a valuable resource for researchers.

**Audience:**

Yes

**Audience Explanation:**

The authors provide a comprehensive review of FWPs, their properties, and their connections with other models in the existing literature. In addition, they suggest how FWPs might be implemented in the brain. This work would be of interest to multiple audiences: the machine learning community working on efficient sequence models and the emerging neuroAI community bridging machine learning and neuroscience. Given its pedagogical approach and progressive structure, it can serve as an accessible primer for researchers new to FWPs.

**Claims And Evidence:**

Yes

**Claims Explanation:**

The authors mathematically demonstrate the formal connections between FWPs and various classes of models. They provide key facts and intuitions from cited papers in a clear and concise manner.  Furthermore, they comprehensively cite relevant literature throughout the paper, grounding claims in existing work, and they build understanding for readers progressively. While drawing connections to neurobiology, they clearly mention where their claims are speculative.

As a review/primer paper, the work appropriately synthesizes existing research rather than providing new empirical validation, while the mathematical connections remain clear and claims remain well-supported by cited literature.

**Requested Changes:**

- Page 5, Section 2.2: When introducing 'One type of SSM that has recently received much attention in ML,' I would like to request the authors to provide a citation for this specific SSM variant.
- Page 12, Section 3.4: The statement 'while Gated DeltaNet has been reported to outperform DeltaNet on language modeling tasks, consistency of this advantage in other tasks has not been confirmed yet (e.g., we found the opposite trend for reinforcement learning in certain game environments)' appears to lack a citation. I would request the authors to provide a reference for the reinforcement learning results mentioned, or clarify if this is unpublished work.
- Section 4.1: Please add brief background explanations of AMPA and NMDA receptors for readers without neuroscience training. As many members of the machine learning community may be unfamiliar with these neurobiological mechanisms.
- In Table 1, the local objective functions are listed but their derivations for most models are not presented. If possible, it would be helpful to add the derivations for these local objective functions.
- There is a typo on page 17 - "diagnoal".

---

> ### Author Response · Authors · 2025-10-28
> **Author response**
>
> We thank the reviewer for the valuable and thorough comments, as well as for the overall positive feedback on this paper as a primer relevant to both the machine learning and neuroscience communities.
>
> All the requested changes are meaningful, and we will address them in the revision. More specifically:
>
> > Page 5, Section 2.2: When introducing 'One type of SSM that has recently received much attention in ML,' I would like to request the authors to provide a citation for this specific SSM variant.
>
> Thank you for pointing this out. We will change this sentence to "An interesting class of SSM can be obtained by..." as we currently dedicate the two last paragraphs of this section to provide many references with the historical development about this model class (as we cannot reduce this model class to a single citation).
>
> > Page 12, Section 3.4: The statement 'while Gated DeltaNet has been reported to outperform DeltaNet on language modeling tasks, consistency of this advantage in other tasks has not been confirmed yet (e.g., we found the opposite trend for reinforcement learning in certain game environments)' appears to lack a citation. I would request the authors to provide a reference for the reinforcement learning results mentioned, or clarify if this is unpublished work.
>
> Thank you for pointing this out. Reviewer HYzD made a similar comment. We will amend this part accordingly.
>
> > Section 4.1: Please add brief background explanations of AMPA and NMDA receptors for readers without neuroscience training. As many members of the machine learning community may be unfamiliar with these neurobiological mechanisms.
>
> This is an excellent idea. We will add these technical terms from neuroscience to the glossary to make them more accessible.
>
> > In Table 1, the local objective functions are listed but their derivations for most models are not presented. If possible, it would be helpful to add the derivations for these local objective functions.
>
> While we will not include full derivations for all models, we will add a box providing derivations for some of the representative examples.
>
> > There is a typo on page 17 - "diagnoal".
>
> Thank you for catching this. We will correct it in the revision.
>
> We hope these fixes will fully addresses your remaining concerns. Thank you again for all your suggestions.

---

### Author Response · Authors · 2025-10-28
**General response**

We thank all the reviewers for their valuable feedback on our submission.
We are glad to hear that the reviewers found our work relevant overall and that no major issues remain to be resolved.

Please find our individual responses to each reviewer below.

We will incorporate all of the reviewers’ suggestions into the manuscript and upload the revised PDF within the next couple of days. We will notify you once it has been uploaded. If you have any additional suggestions in the meantime, please let us know.

---

### Author Response · Authors · 2025-11-05
**Revision Uploaded**

Dear Reviewers,

We have uploaded the revised PDF.

As there were no objections to the edit proposals in our responses addressing the reviewers’ comments, the revision follows the corresponding descriptions (please refer to the **Change Summary** list above). Please let us know if there are any further edit requests.

Once again, we sincerely thank all the reviewers for their valuable feedback and constructive suggestions, which have helped us improve the manuscript.

---

### Author Response · Authors · 2025-11-11
**End of discussion period, and thank you**

As the discussion period is coming to a close, we assume that the reviewers who have not provided further comments on our response are also satisfied with our revisions.

We thank all the reviewers once again for their valuable feedback.

---

### Decision · Action_Editor_k478 · 2025-12-09

**Recommendation:** Accept with minor revision

**Additional Comments:**

Please include a discussion of the following relevant literature:
Limbacher, T., & Legenstein, R. (2020). H-mem: Harnessing synaptic plasticity with hebbian memory networks. Advances in Neural Information Processing Systems, 33, 21627-21637.
Limbacher, T., Özdenizci, O., & Legenstein, R. (2023). Memory-dependent computation and learning in spiking neural networks through Hebbian plasticity. IEEE Transactions on Neural Networks and Learning Systems.
Aitken, K., & Mihalas, S. (2023). Neural population dynamics of computing with synaptic modulations. Elife, 12, e83035.

These articles are in particular relevant from the (computational) neuroscience perspective.

**Audience:**

Yes

**Audience Explanation:**

A diverse subset of the TMLR's audience will be interested in the paper, including researchers that are interested in brain-inspired machine learning but also those that are in general interested in deep learning.

**Claims And Evidence:**

Yes

**Claims Explanation:**

The manuscript provides a review over a class of neural networks known as "fast weight programmers". The overview is to a large extent comprehensive and exhibits important relations to other neural architectures (transformers).

---

> ### Author Response · Authors · 2025-12-31
> **Camera-ready uploaded and thank you**
>
> Dear Action Editor and Reviewers,
>
> Thank you for your favorable final decision. We have uploaded the camera-ready version, incorporating the final changes requested by the Action Editor.
>
> We sincerely appreciate the time and effort you invested in handling and reviewing our paper.
>
> Thank you very much once again, and best wishes for the New Year.
>
> Best regards,
>
> Authors